# The circadian oscillator analysed at the single-transcript level

Nicholas E Phillips[1],† (iD), Alice Hugues[1,2],†, Jake Yeung[1] (iD), Eric Durandau[1] (iD), Damien Nicolas[1] & Felix Naef[1],* (iD)

## Abstract

**The circadian clock is an endogenous and self-sustained oscillator that anticipates daily environmental cycles. While rhythmic gene expression of circadian genes is well-described in populations of cells, the single-cell mRNA dynamics of multiple core clock genes remain largely unknown. Here we use single-molecule fluorescence in situ hybridisation (smFISH) at multiple time points to measure pairs of core clock transcripts, *Rev-erbα* (*Nr1d1*), *Cry1* and *Bmal1*, in mouse fibroblasts. The mean mRNA level oscillates over 24 h for all three genes, but mRNA numbers show considerable spread between cells. We develop a probabilistic model for multivariate mRNA counts using mixtures of negative binomials, which accounts for transcriptional bursting, circadian time and cell-to-cell heterogeneity, notably in cell size. Decomposing the mRNA variability into distinct noise sources shows that clock time contributes a small fraction of the total variability in mRNA number between cells. Thus, our results highlight the intrinsic biological challenges in estimating circadian phase from single-cell mRNA counts and suggest that circadian phase in single cells is encoded post-transcriptionally.**

**Keywords** circadian oscillator; single cells; smFISH; stochastic gene expression; transcriptional bursting

**Subject Categories** Chromatin, Transcription & Genomics; RNA Biology

**Mol Syst Biol. (2021) 17: e10135**

## Introduction

In animals, the circadian clock is a 24-h period oscillator that dynamically regulates central aspects of physiology across all scales of biological organisation, from metabolism and locomotor activity down to cellular gene expression and cell signalling (Mohawk *et al*, 2012; Yeung & Naef, 2018; Reinke & Asher, 2019). Circadian rhythms are entrained by external signals (Zeitgebers) but are present even in single cells; in fact, live-cell imaging of transcriptional reporters (Nagoshi *et al*, 2004) and endogenous fusion proteins (Welsh *et al*, 2004; Leise *et al*, 2012; Li *et al*, 2020) have revealed that circadian oscillations are cell-autonomous and self-sustained, with a period that fluctuates from cycle to cycle by typically 10%. Molecularly, it is thought that the oscillations involve interactions between core circadian clock proteins including BMAL1, CRY1 and REV-ERBα (encoded *by Arntl, Cry1,* and *Nr1d1* genes, respectively), establishing negative feedback loops (Takahashi, 2017). Live-cell studies of circadian oscillators in individual cells have thus far remained limited to one gene product at a time, and hence the properties of circadian oscillators in single cells across multiple genes remain largely uncharacterised (Nagoshi *et al*, 2004; Welsh *et al*, 2004).

We thus aimed at characterising circadian oscillators at the transcript level by using multichannel mRNA smFISH, which is a sensitive measure of single-cell transcript counts for multiple genes simultaneously (Raj *et al*, 2008; Itzkovitz & van Oudenaarden, 2011). While such smFISH measurements provide transcript counts at a single snapshot, the resulting mRNA distributions contain rich information about the underlying dynamical processes. Specifically, the mRNA distributions can be analysed using the telegraph model of transcription (Peccoud & Ycart, 1995; Munsky *et al*, 2012) to estimate the transcriptional bursting kinetics of individual genes using either smFISH (Raj *et al*, 2006; Gómez-Schiavon *et al*, 2017; Nicolas *et al*, 2018; Zoller *et al*, 2018; Mermet *et al*, 2018) or even scRNA-seq count data (Kim & Marioni, 2013; Bahar Halpern *et al*, 2015; Larsson *et al*, 2019), though the latter is less sensitive. At steady state, the distribution predicted by the model, a Beta-Poisson mixture (Kim & Marioni, 2013; Dattani & Barahona, 2016), can be approximated with a negative binomial (NB) distribution, which is valid when the mRNA half-life is long in relation to the time spent in the active promoter state (Raj *et al*, 2006) and which is typical for mammalian genes (Suter *et al*, 2011; Zoller *et al*, 2015). The NB distribution, which is over-dispersed (having larger variance than expected from a Poisson distribution), uses two informative parameters specifying its shape: the burst size and the burst frequency (normalised by mRNA half-life). Using the telegraph model, the transcriptional parameters for the core clock gene *Bmal1* have been analysed both by smFISH (Nicolas *et al*, 2018) and live imaging of destabilised *Bmal1-Luc* transcriptional reporters (Suter *et al*, 2011;

1   Institute of Bioengineering, School of Life Sciences, Ecole Polytechnique Fédérale de Lausanne, Lausanne, Switzerland
2   Master de Biologie, École Normale Supérieure de Lyon, Université Claude Bernard Lyon I, Université de Lyon, Lyon, France
    *Corresponding author. Tel: +41 21 693 16 21; E-mail: felix.naef@epfl.ch
    †These authors contributed equally to this work

   *Molecular Systems Biology*   17: e10135 | 2021   **1 of 18**

Zoller *et al*, 2015). These studies showed that the transcriptional burst frequencies of *Bmal1* and the clock-output gene *Dbp* oscillate over the circadian clock (Nicolas *et al*, 2018), while the burst size stays constant.

In addition to gene expression variability caused by biomolecular birth–death processes and transcriptional bursting, both of which contribute intrinsic noise, significant differences in mRNA number may also be caused by extrinsic sources of cell-to-cell variability (Elowitz *et al*, 2002; Zechner *et al*, 2014), such as cell size or cell cycle stage (Battich *et al*, 2015; Bruggeman & Teusink, 2018; Foreman & Wollman, 2020). The scaling of mRNA number with cell size has been attributed to differences in transcriptional activity, although not all genes are subjected to cell-size control (Schmidt & Schibler, 1995). Notably, transcriptional burst size has been found to correlate with cell volume in both mammalian cells (Padovan-Merhar *et al*, 2015) and yeast (Sun *et al*, 2020), i.e. transcriptional burst sizes are larger in larger cells. The molecular mechanism that underpins this scaling has been proposed to integrate both cellular volume and DNA content in order to produce the appropriate amount of RNA for a cell of a given size, which is consistent with a model whereby a factor limiting for transcription is sequestered to the DNA (Padovan-Merhar *et al*, 2015). For oscillatory transcripts of the circadian clock and also transcripts driven by it, variability in the single-cell state of the circadian oscillator (partial synchronisation of the circadian phases) would also further increase transcript count variability in cell populations (Pulivarthy *et al*, 2007). The extent to which each of these sources of noise contributes to cell-to-cell heterogeneity in mRNA expression remains an open question for circadian clock genes.

Here, we aimed to quantitatively study how the mRNA distributions of core circadian genes evolve over the circadian cycle while considering multiple sources of intrinsic and extrinsic variability. We used smFISH to simultaneously target pairs of core clock transcripts (*Cry1* and *Nr1d1*, or *Cry1* and *Bmal1*) in confluent NIH3T3 mouse fibroblasts every 4 h over the circadian clock. We mathematically modelled the mRNA counts using mixtures of distributions to account for intrinsic transcriptional fluctuations (bursting), cell-to-cell variability and time-varying (periodic) parameters. We investigated several models of increasing complexity and, using a Bayesian model selection approach, we found that the preferred model favours inclusion of measured cellular area as an explanatory variable, with gene-specific scaling of mRNA counts with cell size. From the preferred model, we could decompose the sources of measured variation in core clock transcript number, showing that only a small percentage is caused through circadian time. Due to the strong contribution of transcriptional bursting in individual genes, our results suggest that circadian phase in individual cells may be specified by mRNA numbers of many genes together or by other molecular states such as protein abundance or protein activity levels.

# Results

## Time-resolved mRNA count distributions of core clock genes in single cells

To characterise the circadian oscillator at the single-transcript level, we performed smFISH in synchronised, confluent (non-dividing) NIH3T3 mouse fibroblasts and measured transcript numbers every 4 h for 24 h (7 time points) (Fig 1A). The cells were synchronised using dexamethasone (Dex), and sampling started 17 h after the treatment to avoid the initial transient response. We performed multichannel imaging with fluorescent probes targeting exons of *Bmal1* and *Cry1* or *Nr1d1* and *Cry1* in the same cells. We measured the number of transcripts per cell for each gene in approximately 450 single cells per time point from three replicates (Materials and Methods). After cell segmentation (Appendix Fig S1), the mRNA distributions for each replicate are shown in Appendix Figs S2–S5. The mean number of *Bmal1*, *Cry1* and *Nr1d1* transcripts per cell oscillated along the circadian cycle and ranged from 10 to 35 molecules (Fig 1B). *Bmal1* mRNA counts were in the same range as previous measurements in the same cell line (Nicolas *et al*, 2018). To estimate the population phase and amplitude of each gene, we fitted a two-harmonic cosinor model (Materials and Methods). The oscillations of *Cry1* and *Bmal1* are approximately in antiphase with *Nr1d1* positioned in-between, which is consistent with known phase relationships of core clock transcripts in 3T3 cells in bulk measurements (Hughes *et al*, 2009; Ukai-Tadenuma *et al*, 2011) (Fig 1C). The fold change, defined as the ratio of the peak to the trough, was 1.5 for *Cry1*, 1.6 for *Bmal1* and 2.1 for *Nr1d1* (Fig 1C). Even though the cells were synchronised with Dex, we still expect cell-to-cell differences in the phase resulting from incomplete synchronisation (Pulivarthy *et al*, 2007). As a comparison, we entrained cells with temperatures cycles, known to be an efficient method of synchronisation, and obtained a fold change of 2.1 for *Bmal1*, which was slightly higher than with Dex synchronisation and similar to

**Figure 1. Single-molecule RNA fluorescence in situ hybridisation (smRNA FISH) captures transcript distributions of core clock genes in mouse fibroblasts at multiple time points.**

A  smFISH targeting *Bmal1*, *Cry1* and *Nr1d1* in wild-type NIH 3T3 cells at 17, 29 and 25 h after synchronisation with Dex, respectively. Nuclei are stained with DAPI (blue). Each fluorescent dot (white) corresponds to a single transcript. Segmented cell boundaries are delineated in grey. The blue and white channels represent maximum z-projections. Scale bar: 20 μm.

B  Number of *Bmal1*, *Cry1*, and *Nr1d1* transcripts per cell as a function of time after treatment with Dex. These data combine all smFISH hybridisations from the 4 h sampled *Nr1d1/Cry1* and *Bmal1/Cry1* experiment. Each dot shows the average over one replicate from an independent slide (Materials and Methods). The solid lines represent fits using a two-harmonic cosinor model (Equation (1), Materials and Methods) for each gene individually.

C  The inferred peak phase (angular component of the graph) and max-to-min fold change (radial component) from the fit of Equation (1).

D  Distributions of *Nr1d1*, *Cry1* and *Bmal1* transcripts at 21, 29 and 37 h after synchronisation with Dex. $\mu$ represents the mean of the distribution, CV represent the coefficient of variation (standard deviation / mean), and *N* is the number of cells at the given time point. Total number of cells analysed for the *Nr1d1/Cry1* pair: 21 h—449; 25 h—414; 29 h—521; 33 h—463; 37 h—477; 41 h—429. Total number of cells analysed for the *Bmal1/Cry1* pair: 17 h—465; 21 h—490; 25 h—504; 29 h—436; 33 h—407; 37 h—454; 41 h—404.

Source data are available online for this figure.

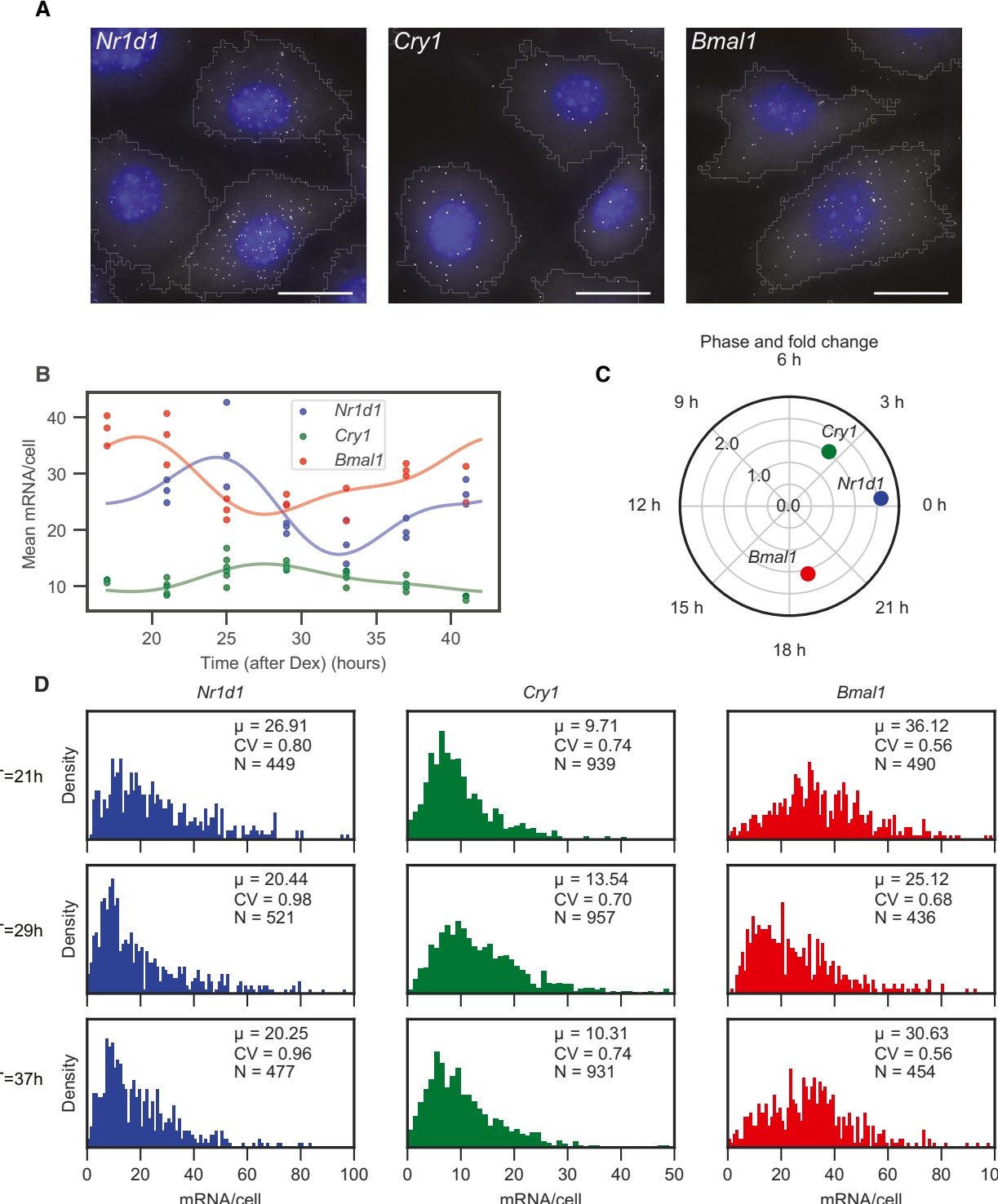

**Figure 1.**

previous reports (Saini *et al*, 2012) (Appendix Fig S6). To demonstrate the reproducibility of the temporal mRNA patterns, we additionally performed an independent experiment on four time points for *Nr1d1* and *Cry1*. This experiment showed that the oscillatory patterns are reproducible (Appendix Fig S7).

While the mean mRNA level changed over time for all genes, there was substantial variability in mRNA numbers between cells,

leading to significant overlap of the transcript distribution between different time points (Fig 1D). The measured coefficients of variation (CV = standard deviation/mean) ranged from 0.5 to 1. For comparison, the expected CVs would be 0.17–0.32 if a gene had an average mRNA level between 10–35 molecules and if mRNA dynamics followed a Poisson process (i.e. assuming constant mRNA production and degradation rates and that cells share the same

kinetic parameters). Thus, the observed variability implies additional sources of variability, possibly from transcriptional bursting or extrinsic sources (Battich *et al*, 2015). The mathematical modelling introduced below serves to identify these contributions.

## Core clock transcript numbers scale with cell size

Given that previous studies identified cell size as a source of extrinsic variability in transcript counts (Battich *et al*, 2015; Kempe *et al*, 2015; Padovan-Merhar *et al*, 2015; Foreman & Wollman, 2020), we quantified cell sizes and found a positive correlation with mRNA number for all genes (Appendix Fig S8). In fact, we found a linear relationship between the log area and the log mean mRNA number, indicating a power-law scaling (Fig 2A). The relationship was sublinear, with exponents ranging from 0.42 for *Nr1d1* to 0.70 for *Bmal1*. If mRNA counts were proportional to volume and if the cell had a geometry such as a sphere or cube, we would expect a superlinear scaling between mRNA counts and area. The observed sublinear scaling thus likely reflects the "fried egg" morphology of adherent cells, where increases to the area of a relatively flat cytoplasm have a proportionally small effect on the total volume due to the nucleus.

## Two-gene mRNA count distributions reveal gene-pair specific correlations

We next exploited the ability of smFISH to measure multiple transcripts simultaneously to explore the joint relationship between transcript numbers of different clock genes. Our dual-channel imaging allows either *Bmal1/Cry1* or *Nr1d1/Cry1* to be measured in the same cells (Fig 2B). The bivariate relationships between the gene pairs show that *Bmal1/Cry1* are positively correlated at each time point (R from 0.12 to 0.53), whereas *Nr1d1/Cry1* show negative correlations (R from 0.0 to −0.19) (Fig 2C). To also estimate the correlation between genes while accounting for cell area, we regressed out the area for each gene and recalculated the correlation coefficients (Padovan-Merhar *et al*, 2015; Hansen *et al*, 2018). Since all genes are positively correlated with area (Fig 2A), this processing shifted the correlations for both pairs of genes. Specifically, the correlation coefficients for the area-filtered mRNA counts decreased but remained positive for *Bmal1/Cry1* and became more negative for *Nr1d1/Cry1* (Appendix Fig S9). These residual correlations could be caused by a spread in the circadian phases between cells (Wu *et al*, 2018) or regulatory interactions (e.g. NR1D1 protein represses *Cry1* transcription), which can cause different steady-state correlations depending on whether feedback is negative or positive (Munsky *et al*, 2012). Below, we formulate these hypotheses as simplified, effective mathematical models to quantify the compatibility of our data with both scenarios.

## Modelling periodic mRNA count distributions in heterogeneous cell populations as mixtures of negative binomials

We next developed a model with the aim of finding a compact mathematical representation of the circadian clock in the space of multidimensional transcript counts. In other words, we sought to describe how the multivariate probability distribution of mRNA counts varies as a function of circadian time. As the above

exploratory analysis shows, there are systematic effects (e.g. cell features) coupled with time-varying parameters (the circadian oscillator), and the variance of the mRNA distributions is larger than expected with a Poisson process. To keep the model both manageable and interpretable, we made a number of simplifying assumptions. First, since time scales associated with transcriptional bursting as well as mRNA half-lives of clock genes are short compared to the 24-h period, we modelled the system in a quasi-steady state, which means that at each point the mRNA count distribution in the population is approximated with a slowly time-varying stationary distribution. Second, based on reports that transcriptional bursts are typically short relative to the mRNA half-life (Suter *et al*, 2011; Zoller *et al*, 2015), we used a common approximation of the full telegraph model which takes the form of an NB distribution, with burst size and burst frequency as the two effective parameters (Raj *et al*, 2006). We also proposed additional model features in four different but related models of increasing complexity (Table 1 and Fig 3A). As detailed below, we then selected the optimal model from the candidates using Bayesian model selection.

The first and simplest model (M1) assumes that the transcriptional bursting parameters controlling the shape of the NB distribution are modulated in a deterministic way by the clock, with no further sources of cell-to-cell heterogeneity. Moreover, we incorporate previous findings regarding transcriptional bursting kinetics of clock genes to further reduce the complexity of the model. Namely, previous results from the same cell line have shown that burst frequency of clock genes is modulated by the circadian clock while the burst size remains constant (Nicolas *et al*, 2018). In M1, we therefore used a NB distribution with an oscillatory burst frequency but otherwise constant parameters (features F1 and F2, Table 1 and Fig 3A).

The second model explicitly incorporates cell size as a source of extrinsic variability. Previous work in mammalian cells showed that transcriptional burst sizes scales with cell volume (Padovan-Merhar *et al*, 2015). In model M2, we allowed the burst size to scale with cell size, and the full mRNA distribution across the population at each time point consequently becomes a mixture of NB distributions (feature F3, Fig 3A). Since we measured cellular area instead of volume, we set the dependence between burst size and cell area using an additional, gene-specific exponent $\beta$, which is also supported by the linear relationship observed between log area and the log mean mRNA (Fig 2A).

For both models M1 and M2, the likelihood of observing the data given the parameters of the model is evaluated using the model-specific NB distribution and the mRNA counts for both genes in each cell. This is performed for both *Bmal1/Cry1* and *Nr1d1/Cry1* pairs across all time points, and this likelihood is combined with model priors to define the posterior parameter distribution for each model (Materials and Methods). We applied Hamiltonian Monte Carlo sampling within the STAN probabilistic programming language to sample the posterior distribution and infer model parameters (Carpenter *et al*, 2017) (parameter estimates for each model shown in Appendix Tables S2–S5). We then quantified the performance of each model by estimating the out-of-sample predictive accuracy, which rewards good fits to the data while penalising model complexity in order to control for overfitting. Specifically, we used approximate leave-one-out cross-validation with Pareto-smoothed importance sampling (PSIS-LOO) to estimate the pointwise out-of-

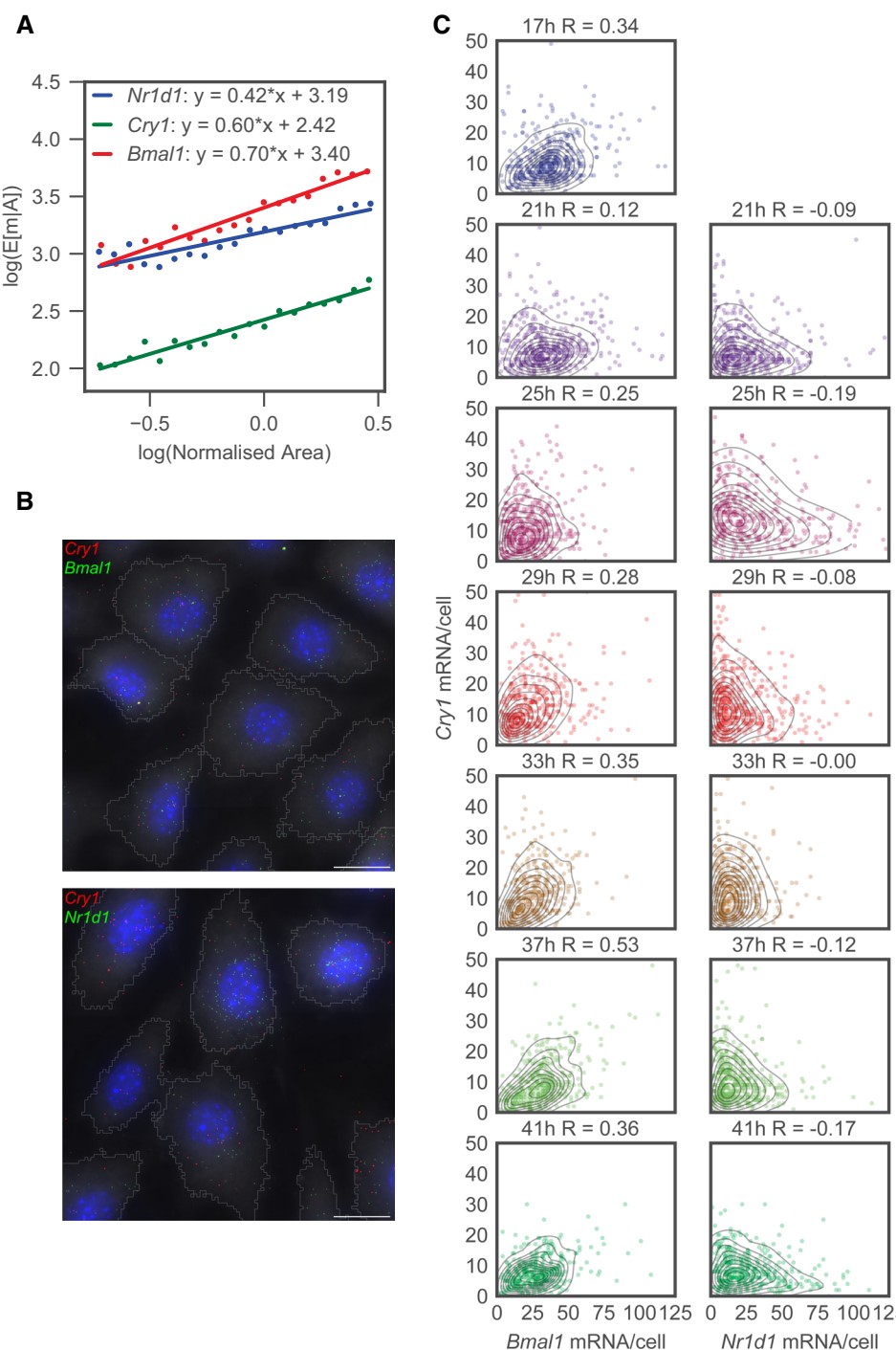

**Figure 2.  Core circadian clock transcript distributions show area dependence and distinct gene-to-gene correlation structures.**

A   Log conditional mean mRNA count for a given area as a function of the log area. Each cell is sorted by area by placing it in one of 20 bins that range from the 5[th] to the 95[th] percentile. The conditional mean given the area E[m|A] is then the mean mRNA count of each bin. The area is normalised such that the average area is equal to one. Solid lines represent a least-squares linear fit to the data.

B   Top image: Dual-channel smFISH targeting *Bmal1* and *Cry1* simultaneously, taken at 17 h after Dex synchronisation. Blue, nuclei stained with DAPI; green dots, *Bmal1* transcripts; red, *Cry1* transcripts. Bottom image: Dual-channel smFISH targeting *Nr1d1* and *Cry1* simultaneously, taken at 25 h after Dex synchronisation. Blue, nuclei stained with DAPI; green dots, *Nr1d1* transcripts; red, *Cry1* transcripts. Scale bar: 20 μm.

C   Bivariate distributions of mRNA counts per cell for dual-channel smFISH targeting either *Bmal1* and *Cry1* or *Nr1d1* and *Cry1*. Each dot corresponds to a single cell, and the contours represent KDE estimates of the density. Time represents the number of hours after Dex synchronisation, and R represents the Pearson correlation coefficient.

**Table 1.  The features (F1-F5) included in each of the four models (M1-M4) considered.**

|  | M1 | M2 | M3 | M4 |
|---|:---:|:---:|:---:|:---:|
| F1: Time-dependent burst frequency | ✓ | ✓ | ✓ | ✓ |
| F2: Constant burst size | ✓ |  |  |  |
| F3: Cell size-dependent burst size |  | ✓ | ✓ | ✓ |
| F4: Cell-specific circadian phase |  |  | ✓ |  |
| F5: Cell-specific burst size |  |  |  | ✓ |

sample prediction accuracy (as described in Vehtari *et al* (2017)), where models with the highest PSIS-LOO score have the best predictive accuracy. The PSIS-LOO calculated for models M1 and M2 showed a clear preference for M2 compared to M1 (Fig 3B), and we hence built subsequent models using cell size-dependent burst sizes. The dependence on size differed between the three genes (parameter $\beta$, Fig 3C), with *Nr1d1* having the weakest and *Bmal1* the strongest dependency, suggesting that the scaling between mRNA number and cell size for the measured clock transcripts can be gene-specific.

To capture additional features observed in the data, notably the positive and negative correlations in Fig 2C and Appendix Fig S9, we also allowed cells to be imperfectly synchronised (model M3). A spread of circadian phases in a cell population could distort the correlation in mRNA counts between genes, as two genes that are on average in-phase/antiphase but where phases vary between cells could generate positive/negative correlations. To model potentially imperfect synchronisation of the cells at each time point, we introduced cell-specific phases that are distributed around a population average (feature F4, Fig 3A). To ensure these cell phases do not become too spread, we used a von Mises prior with $k = 2$, which approximately matches the phase spread observed in live microscopy for mammalian fibroblasts (Pulivarthy *et al*, 2007). To simplify the analysis, we fixed parameters previously contained in model M2 to their posterior mean values, and in M3, we keep the waveform shape and phase for the bursting frequency fixed (from M2) but allow the burst frequency amplitude to vary. Fitting Model M3 is practically more difficult as it involves many local minima. We therefore performed inference multiple (eight) times and chose the posterior chain with the highest average likelihood. Compared to model M1, the amplitude of oscillations in burst frequency increases by approximately 5%, indicating that the underlying amplitude in single cells is greater than observed at the mean level

due to partial synchronisation (Appendix Fig S10); however, the improvement was minor and the increased complexity of M3 over M2 was not supported according to the PSIS-LOO criterion. Thus, it was apparently difficult to use model M3 to correct the individual phase for each cell, likely due to the fact that the two mRNA counts measured in each cell do not contain sufficient phase information and that the global optimisation problem contains many local minima. This could potentially be improved by measuring more genes simultaneously.

Finally, we considered a further refinement (model M4) that introduces variable burst size from cell to cell as an alternative mechanism to explain correlations between gene pairs. While in model M2 the burst size was assumed to be directly proportional to the cell area, in model M4 we included an additional cell-specific random variable $\epsilon_{i,g}$ to modify the burst size in each cell $i$ for each gene $g$ (feature F5, Fig 3A). Model M4 is a hierarchical model whereby the distribution of cell-specific parameters is also learnt during inference, and we assumed that the $\epsilon_{i,g}$ is bivariate log-normally distributed between pairs of genes. This log-normal distribution is parameterised with $\mu_g$, $\sigma_g$ and $\rho$, which represent the mean, variance and correlation of $\epsilon_{i,g}$ between genes (in log space). Practically, this means that when bursts are large for one gene, they can be large (correlated) or small (anticorrelated) for another gene, and while the precise mechanism is left unspecified, it can be interpreted as a signature of regulatory interaction (see below). When all parameters are free, we noticed that the burst frequency can become unrealistically high due to a tendency to overfit to individual cells, and we therefore locked the burst frequency to the posterior mean values from model M2. The PSIS-LOO scores overall favoured model M4 (Fig 3B), and the predicted joint probability density shows good similarity to the observed data (Fig 3D) (all time points shown in Appendix Fig S11). To further validate the predictive performance of each model, we performed a "leave-replicate-out" cross-validation, with the aim of testing how well the predictions of each model generalise to cells that are not in the training set. We trained each model while omitting the data from one gene in a test slide. We then calculated the likelihood score of the test slide using the parameters from the training set, and we repeated this for all slides. Similarly to the PSIS-LOO, the results of the leave-replicate-out cross-validation showed that model M4 has the best predictive performance (Appendix Fig S12).

The inferred average burst frequency was highest for *Bmal1* and lowest for *Nr1d1,* while *Nr1d1* showed the highest and *Cry1* the

**Figure 3.  A mathematical model that includes area dependence and gene-specific correlation in burst size captures the observed time-dependent bivariate mRNA distributions.** ▶

A   Schematic of the five different model features (F1-F5) used in the four models considered (see Table 1). F1: assumes the burst frequency oscillates over 24 h; F2: assumes a fixed burst size for each gene. The burst frequency ($f$) and burst size ($b$) determine the shape of the NB distribution used to model the mRNA counts (see Materials and Methods), where the mean $u = bf$ and the variance $\sigma^2 = \mu + \mu^2/f$. F3: the burst size in each cell scales with measured cell area using a gene-specific exponent $\beta_g$. F4: to model incomplete synchronisation, each cell has a cell-specific phase offset compared to the average phase. F5: the burst size in each cell is proportional to the cell area multiplied by a cell-specific burst parameter $\epsilon_{i,g}$. These cell-specific $\epsilon_{i,g}$ are further modelled using a bivariate log-normal distribution.

B   PSIS-LOO calculated for each of the four models using the whole dataset of both gene pairs across all time points.

C   Posterior parameter estimates. Burst frequency is plotted as a function of time, where the solid line represents the posterior mean and the shaded area represents the 90% confidence interval. Posterior probability densities are shown for the average burst size, $\beta$ (which controls the dependence between burst size and cell area) and $\rho$ (representing the correlation in burst size between genes (in log space)), where $\beta$ is inferred from model M2 and the average burst size and $\rho$ are calculated from model M4.

D   Comparison of the probability density of the data (kernel density estimates) with the model (using model M4). To estimate the probability from the model, the dataset was simulated 15 times using the posterior mean parameter values combined with the measured cell areas.

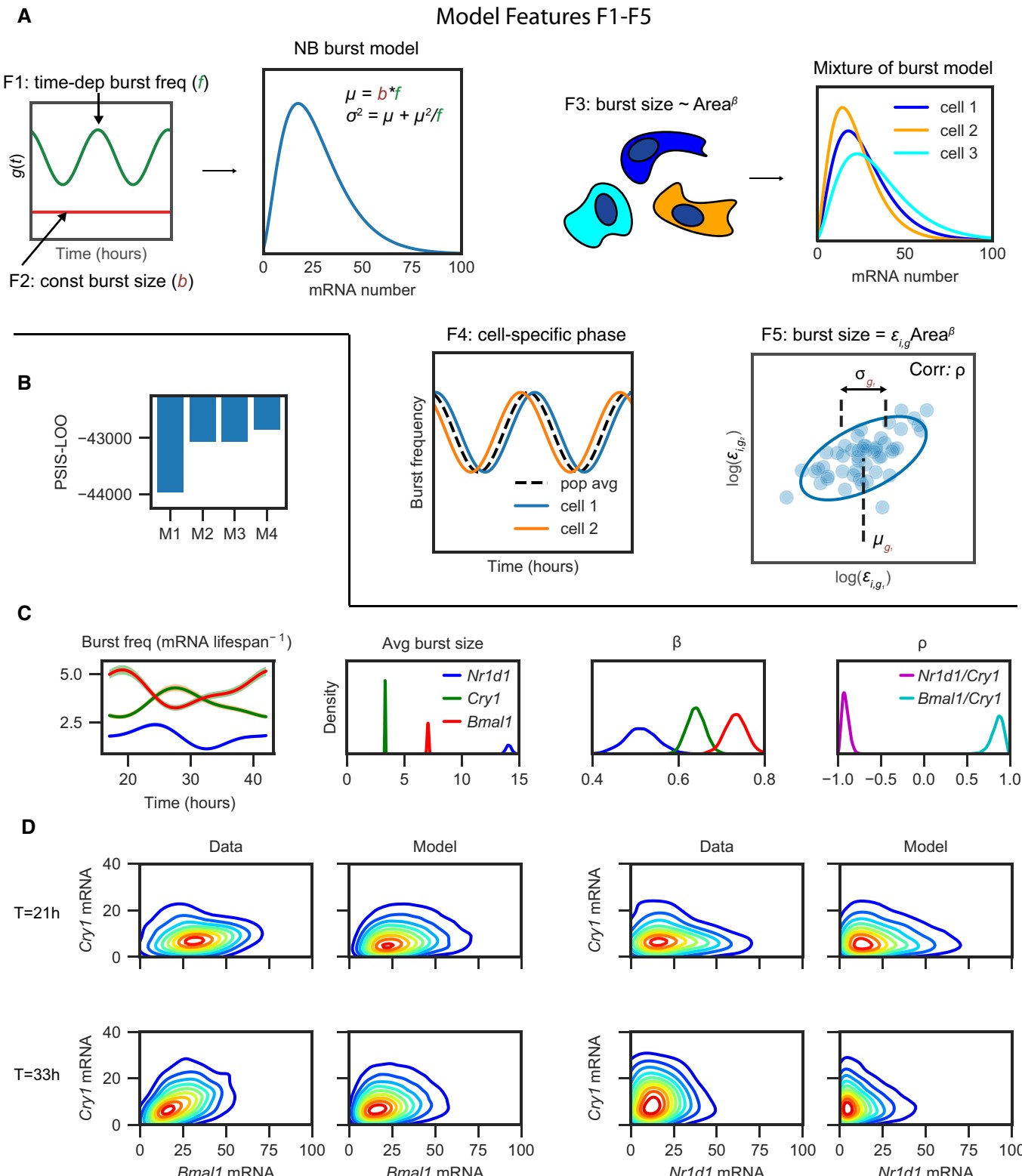

**Figure 3.**

lowest average burst size (Fig 3C). The burst frequencies (normalised by mRNA half-life) were in the upper range when compared to the transcriptome-wide estimates using single-cell RNA-seq, while the burst sizes spanned the observed range

(Larsson *et al*, 2019). The correlation parameter $\rho$ of the random variable $\epsilon_{i,g}$ was positive for *Cry1/Bmal1* and negative for *Cry1/Nr1d1* (Fig 3C), consistent with the observed correlations in the data (Fig 2C and Appendix Fig S9). Overall, we identified a

preferred model (M4) that incorporated transcriptional bursting, cell size and circadian time variation and which was able to capture the measured bivariate smFISH densities.

## Generating gene–gene-specific correlations from the core clock network topology

The preferred model (M4) contains cell-specific burst parameters ($\epsilon_{i,g}$) and a correlation parameter ($\rho$) without explicitly describing their biological origins. Thus, we next investigated whether a dynamic model with gene–gene interactions could provide an interpretation. Creating a detailed model of the circadian oscillator to interpret data is challenging since certain variables (such as protein levels), parameters (such as mRNA/protein half-lives, repression strengths) and additional genes (such as *Clock*) are not directly measured in the experiments. Consequently, we created a simplified representation to explore the connection between the core clock network topology and the preferred model M4. We used the telegraph model of stochastic gene transcription as the basis for the model, where the promoter switches randomly between an active and inactive state (Peccoud & Ycart, 1995) (Appendix Fig S13). The gene state, mRNA and protein levels are modelled for each gene, and to keep the model close to model M4, we used oscillatory functions for the burst frequency while the transcription rate (and hence the burst size) for each gene is a function of the protein levels of the other genes in the network. We used the circadian clock gene network topology for *Nr1d1*, *Cry1* and *Bmal1* as modelled in Relógio *et al* (2011) (the model is fully detailed in Materials and Methods). When the feedbacks are such that the negative repression of *Nr1d1* by CRY1 is high, the network can generate positive mRNA correlation between *Bmal1/Cry1* and negative correlation between *Nr1d1/Cry1* (Appendix Fig S13), as observed in our data (Fig 2C and Appendix Fig S9). Furthermore, using the same inference framework as for our data on the simulated mRNA distributions, the obtained $\rho$ parameter is positive for *Bmal1/Cry1* and negative for *Nr1d1/Cry1*, which was also found for our data (Fig 3C). This analysis therefore supports the hypothesis that gene–gene interactions represent a plausible mechanism for generating correlated burst parameters between genes, which are a feature of the preferred model M4.

The underlying stochasticity of the biochemical reactions implies differences in phase between cells, which M3 should be able to capture, though in practice performing the inference proved difficult. While phase differences undoubtably exist between cells, analysing the consequences on the correlation structure shows that it is not sufficient to explain the observed bivariate mRNA distributions. Indeed, if two genes oscillate with a similar phase, the expression of the two genes will be positively correlated (Appendix Fig S14). Similarly, when the oscillations are in antiphase, negative correlations are found. Given that *Nr1d1* and *Cry1* are closer in phase than *Bmal1* and *Cry1*, one would expect that the correlation between *Nr1d1* and *Cry1* (once accounting for area) would be higher than for *Bmal1* and *Cry1*, which was not found in the data (area-corrected correlations in Appendix Fig S9). It is therefore unlikely that phase noise alone induces the observed correlations, and the simulations of the gene network show that gene–gene interactions also contribute.

## Circadian oscillations at the transcript level are blurred by other sources of single-cell heterogeneity

Having selected a preferred model to describe the data, we next analysed the model to decompose the variance into distinct sources. In model M4, there are four sources of transcript count variability across populations of individual cells: temporal control by the circadian clock, intrinsic noise due to the mRNA production-decay process, extrinsic noise from variable cell size and fluctuations in burst size ($\epsilon_{i,g}$, termed "other extrinsic"). Intrinsic noise can be further partitioned into a Poisson component that arises due to the discreteness of mRNA counts (and would be present with constant, constitutive expression) and a second component caused by transcriptional bursting. To decompose the variance, we use the law of total variance (Materials and Methods).

For all three genes, we found that the variance was dominated by intrinsic noise (Fig 4A), with bursting having a significantly larger contribution than the Poisson component. The extrinsic sources of noise in our model, which include area and variable burst size, contributed less for all three clock genes, and the fraction of variance due to cell area ranged from 4.2% for *Nr1d1* to 17.6% for *Bmal1*. Previous studies have shown that the relative proportions of intrinsic versus extrinsic noise are both condition- (cell types, cell states) and gene-specific, and regression models that use cellular features as explanatory variables can account for between 10 and 80% of the variance, depending on the gene (Battich *et al*, 2015; Foreman & Wollman, 2020). One explanation for the low intrinsic variance in these studies is that transcriptional fluctuations are filtered by nuclear retention, though other reports suggest that Fano factors (variance/mean, a measure of overdispersion compared to the Poisson distribution) can be even larger in the cytoplasm than in the nucleus (Hansen *et al*, 2018). In the cells used here, the strong signature of transcriptional bursting and high intrinsic noise is consistent with live imaging of a *Bmal1* transcriptional reporter in the same cell line under similar growth conditions, where intrinsic noise was estimated to be 4 times larger than extrinsic noise (Zoller *et al*, 2015). As we are only able to quantify the role of extrinsic sources that are included in the model, it is nevertheless possible that inclusion of additional sources such as the cell cycle, phenotypic heterogeneity and microenvironment would increase the proportion of extrinsic noise. Nonetheless, as shown previously (Nicolas *et al*, 2018), cells were maintained in conditions that minimise cell cycle progression and NIH3T3 fibroblasts are phenotypically homogenous, which controls at least for those major sources of extrinsic noise (Gut *et al*, 2015; Foreman & Wollman, 2020).

Somewhat surprisingly, the contribution of the 24-h cycle to mRNA variance was low across all genes, consistent with the large overlap in mRNA distributions at different time points (Fig 1D). This may be unexpected given that robust oscillations in mRNA levels are found in both bulk RNA sequencing in tissues (Zhang *et al*, 2014) as well as cell culture (Hughes *et al*, 2009; Aguilar-Arnal *et al*, 2013). To understand whether our results are compatible with circadian rhythms found in bulk cell line RNA sequencing, we calculated the percentage of variance due to time when increasingly large pools of simulated cells are considered (Fig 4B). Starting from a small percentage with one cell (as previously shown in Fig 4A), the percentage of variance explained by time increases to almost 100% with 1,000 cells, which is caused by progressive averaging-out of

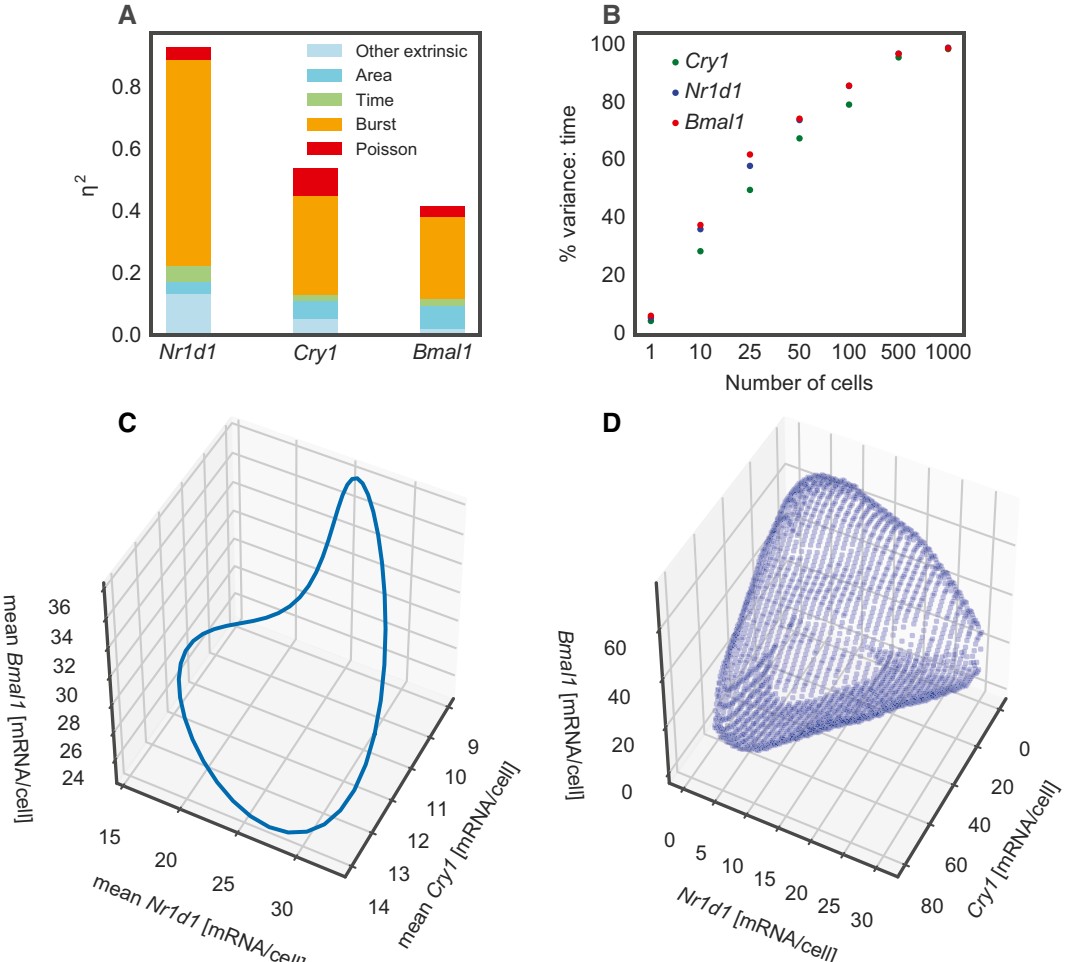

**Figure 4. Decomposing the sources of noise shows time has a small contribution to overall mRNA count variance.**

A   Noise decomposition of model M4, estimated with 100 simulations of the dataset using the posterior mean parameter values combined with the measured cell areas. The variable $\eta^2$ represents the variance/mean$^2$.
B   Simulations from model M4 to show the percentage of variance attributed to time as a function of the number of cells in each sample.
C   Simulation of the 3-dimensional mean mRNA trajectory of *Bmal1*, *Nr1d1* and *Cry1*. Each point on the cycle represents a different circadian time.
D   Simulation of the 3-dimensional probability distribution of *Bmal1*, *Nr1d1* and *Cry1*, averaged over time. The surface represents the area of highest probability density that integrates to 80% of the total probability.

other sources of single-cell noise as cell numbers become larger. Our results on single cells are therefore compatible with RNA sequencing of bulk populations of cells.

While our experiments and inference were performed on pairs of genes, the model allows us to simulate the count distributions for all three genes simultaneously. As the correlation of $\epsilon_{i,g}$ between *Nr1d1* and *Bmal1* is not inferred directly in our model, we use the *Cry1*/*Bmal1* and *Cry1*/*Nr1d1* correlations together with the minimal assumption of conditional independence of *Bmal1* and *Nr1d1* given *Cry1* (Materials and Methods). The three-dimensional simulations help visualise the differences between the average and the single-cell measurement of circadian clock mRNA expression. The mean mRNA level shows a clear periodic structure (Fig 4C), where each circadian time represents a different point on the cycle. In a population of unsynchronised single cells, one might expect noise to blur this structure. It transpires that due to the large contribution of

transcriptional burst noise and the weak contribution of time to mRNA variance, the single-cell joint three-dimensional probability distribution is significantly blurred with a diffuse and amorphous structure (Fig 4D). In summary, probabilistic modelling of the 3-dimensional core clock system shows a 24-h cycle in the mean level, but this well-defined periodic shape is blurry when the full probability density of mRNA counts is considered.

## Discussion

Single-cell measurement of gene expression with smFISH goes beyond bulk mRNA measurement as it allows for the full characterisation of transcript distributions in populations of cells. Until now, the temporal evolution of circadian mRNA distributions through the clock remained uncharacterised. Here we performed a bivariate

measurement of mRNA number across two pairs of genes every 4 h over the circadian clock. There were several notable features of our data including: an average mRNA expression level that oscillated over the circadian clock, a coefficient of variation for each gene that largely exceeded that from a Poisson process, a transcript number that correlated with cell area and a correlation between genes that depended on the pair considered. To capture these phenomena in a unified quantitative framework, we proposed several probabilistic, generative models of our data and used model selection to balance goodness of fit and model complexity to identify the optimal model.

While we have developed this approach for the circadian clock, i.e. a temporal system with a periodic structure, we anticipate that the modelling of smFISH developed here, in particular the ability to include and distinguish intrinsic and extrinsic noise sources, will be relevant to other biological systems. Our starting point was to model the number of mRNA molecules in each cell with a time-varying negative binomial distribution and by subsequently incorporating cell area the resulting distribution of mRNA counts across the entire population of cells became a mixture of NBs. While previous works have modelled extrinsic noise using cell-specific parameters (Llamosi et al, 2016; Phillips et al, 2019; Thompson et al, 2020), we here specifically used mixtures of NB distributions to model smFISH data. While cell area affected both genes in the same cell (and hence always induces positive correlation), we also introduced cell-specific latent variables (either cell phase or burst size) that can alter the bivariate probability distributions more flexibly. Given the ease with which these types of models can now be implemented in probabilistic programming languages such as STAN, we anticipate that the use of such latent variable models will be instrumental in describing and quantifying the underlying (but unobserved) biology that influences gene–gene relationships in multivariate single-cell measurements of gene expression.

The considered models for the circadian smFISH data (Table 1) consisted of different mixtures of negative binomial distributions with a burst frequency that changes over time. The final preferred model had a hierarchical structure and included cell-specific burst sizes that were connected at the population level using a bivariate log-normal distribution, which permitted covariance between different genes. The exact mechanistic interpretation is not specified but could possibly arise from regulatory interactions, given that these three clock genes are intertwined in a web of feedback loops (Appendix Fig S13). For example, BMAL1 activates Nr1d1 transcription, CRY1 is known to repress Nr1d1 transcription by inhibiting BMAL1 activity, while NR1D1 directly represses Cry1 transcription, and hence, the observed pair-dependent correlations may be an outcome of this network (Takahashi, 2017). While computational methods have recently been developed to exploit cell-to-cell heterogeneity to infer gene regulatory topologies (Lipinski-Kruszka et al, 2015; Hilfinger et al, 2016; Chan et al, 2017), building explicit mechanistic models of transcription factor networks remains challenging without additional information on protein levels and activities.

Using the preferred model, we decomposed the variance of the mRNA distributions into distinct components and found transcriptional bursting to be the largest contributor. With further measurements of cellular parameters (e.g. cell volume, mitochondria, cell microenvironment, cell cycle stage), one would expect the estimated contribution of extrinsic noise to grow. However, for the cell type used in our study, cell proliferation was minimised, reducing at least

one major source of extrinsic noise (Nicolas et al, 2018), and live-cell imaging of Bmal1 transcriptional reporter indicated that intrinsic noise was dominant (Zoller et al, 2015). Temporal changes contributed only a small fraction to the total variability in mRNA counts, which is also reflected in the relatively low oscillatory amplitudes we found (Fig 1B). Though cells that were subjected to temperature entrainment did not yield significantly larger amplitudes (Appendix Fig S6), one potential cause we have explored in our modelling (M3) is partial synchrony of our cells. The outcome was that partial synchrony was not the preferred explanation of the low amplitudes, though it could be that we have not found the true minimum of M3 due to too many local minima. Nevertheless, with CVs between 0.5 and 1, which is typical for mammalian genes expressed on the order of 10 molecules (Battich et al, 2015; Zoller et al, 2015), even an increased amplitude of 4-fold would represent only ~38% (CV = 0.5) or ~12% (CV = 1) of the total variance. In other terms, unless the amplitude fold change is radically increased or the CV decreased, time is unlikely to yield a dominant contribution to the total variance.

In principle, the possibility of using transcript counts to estimate single-cell circadian phase is attractive and would have important applications in single-cell genomics; however, the fact that time variability is dominated by other sources of variation highlights significant challenges. A possible solution is to leverage mRNA numbers for a large number of genes, for example using single-cell RNA-seq, as successfully done to estimate cell cycle phase (Liu et al, 2017; Ahmed et al, 2019; Campbell & Yau, 2019). In comparison, analogous methods to estimate circadian phase in single cells currently remain unexplored, although we note though that cell cycle regulated transcripts are expressed at higher levels compared to core clock transcription factors.

The fact that circadian time captures only a low contribution to total mRNA variability may be surprising given that clear circadian oscillations of a single-cell reporter (Rev-Erbα-YFP) are observed in the same NIH 3T3 cells (Nagoshi et al, 2004; Bieler et al, 2014) and in human U2OS cells (Droin et al, 2019). It should be noted that the reporter used in these cells is probably present in many copies, since the cells are typically selected for high signals. The multiple promoters could thus resultantly cause an averaging effect, similar to shown in Fig 4B. Nonetheless, 24-h rhythms are seen in primary fibroblasts dissociated from mPer2-luciferase protein fusion knock-in mice, which shows that robust oscillations are possible at the protein level even when transcribed from diploid alleles. One possible explanation is that there is noise reduction at the protein level. Indeed, using a Bmal1 transcriptional reporter it was shown that the relative amplitude of oscillations at the luminescent reporter protein level was greater than that observed at the mRNA level (Nicolas et al, 2018), which could be caused by regulation of translation or protein degradation over the circadian clock (as known for core clock genes) (Suter et al, 2011; Yoo et al, 2013). Mass spectrometry has demonstrated that many mRNAs with flat mRNA profiles over the circadian clock exhibit oscillatory protein expression (Robles et al, 2014; Mauvoisin et al, 2014; Wang et al, 2017), which could be due to the clock regulated translation (Jouffe et al, 2013) or degradation. Protein degradation via autophagy can be circadian in mouse liver (Ma et al, 2011), and rhythmically regulated protein half-lives have been shown to be responsible for phase differences observed between mRNA and protein levels (Reddy et al, 2006;

Lück *et al*, 2014). In addition to controlling the amplitude of oscillations, the protein stability will also affect the cell-to-cell variability at the protein level, as a more stable protein acts to buffer against temporal fluctuations in mRNA (Raj *et al*, 2006).

Another important aspect of this question is how circadian phase is encoded at the single-cell level, or, in other words, what are the state variables of the clock. For instance, most of the core clock protein functions seem to converge on regulating the rhythmic activity of the CLOCK-BMAL1 transcription factor complex, thus integrating multiple layers of regulation such as protein stability and dimerisation, nuclear import, binding of co-factors and repressors and phosphorylation of DNA binding domains (Partch *et al*, 2014). The process of molecular complex assembly could theoretically alter the noise properties of the circadian state variable, and simple mathematical models have shown that the magnitude of fluctuations of molecular complexes can be lower than their constituent species (Konkoli, 2010). Our data suggest that endogenous gene readouts at the transcript count level are too noisy to faithfully encode single-cell phase, which is however likely defined as a systems property involving activities of protein complexes that are subject to regulation at the post-transcriptional and post-translational levels (Cheong & Virshup, 2011; Partch *et al*, 2014; Aryal *et al*, 2017). In such a complex system, the noise is a function of the network topology (Raj *et al*, 2010), and interactions between components means that noise at the mRNA level is not guaranteed to propagate downstream. In the c-fos/c-jun gene-regulatory pathway, for example, joint measurement of mRNA and protein variability has revealed a noise bottleneck whereby the heterodimers of c-fos and c-jun proteins are less variable than the encoding mRNAs (Shah & Tyagi, 2013). In that system, the response of downstream genes is also affected by a chromatin context that protects against fluctuations in the heterodimer, and it is possible that clock-dependent genes also perform noise filtering in a circadian context.

In sum, our results show that mRNA count distributions of the core clock genes *Bmal1*, *Nr1d1* and *Cry1* are subjected to significant variability such that the circadian limit cycle attractor is blurred at the transcript count level, and future studies may ascertain how single cells are successfully able to generate robust oscillations that have been observed in live-cell imaging.

# Materials and Methods

### Reagents and Tools table

| Reagent/resource | Reference or source | Identifier or catalogue number |
|---|---|---|
| **Experimental models** | | |
| NIH3T3 (*M. musculus*) | Ueli Schibler (Morf *et al*, 2012) | |
| **Oligonucleotides and sequence-based reagents** | | |
| smFISH exonic probes *Bmal1* | This study | Table EV1 |
| smFISH exonic probes *Nr1d1* | This study | Table EV1 |
| smFISH exonic probes *Cry1* | This study | Table EV1 |
| **Chemicals, enzymes and other reagents** | | |
| DMEM | Thermo Fisher Scientific | 11965092 |
| Foetal Bovine Serum (FBS) | Sigma-Aldrich | F7524 |
| Penicillin–Streptomycin–Glutamine (PSG) antibiotics | Thermo Fisher Scientific | 10378016 |
| Trypsin | Thermo Fisher Scientific | 12604013 |
| Fibronectin | Sigma-Aldrich | F0895 |
| Phosphate-buffered Saline (PBS) | Thermo Fisher Scientific | 10010023 |
| Dexamethasone | Sigma-Aldrich | D4902 |
| Coverslips | Marienfeld Superior | 0111580 |
| Cell culture plate, 12-well | Greiner Bio-One | 665 180 |
| Formaldehyde | Sigma-Aldrich | F15587 |
| Stellaris Wash Buffer A | Stellaris | SMF-WA1-60 |
| Nuclease-free water | Thermo Fisher Scientific | AM9932 |
| Deionised Formamide | PanReac AppliChem | A2156 |
| Hybridisation Buffer | Stellaris | SMF-HB1-10 |
| Ribonucleoside Vanadyl Complex | New England Biolabs | S1402S |
| Yeast tRNA | Ambion | AM7119 |
| DAPI | Thermo Fisher Scientific | D1306 |
| HCS CellMask Green Stain | Thermo Fisher Scientific | H32714 |

Reagents and Tools table (continued)

| Reagent/resource | Reference or source | Identifier or catalogue number |
|---|---|---|
| Stellaris Wash Buffer B | Stellaris | SMF-WB1-20 |
| ProLong Gold Antifade Mountant | Thermo Fisher Scientific | P10144 |
| **Software** | | |
| CellProfiler3.1.8 | https://cellprofiler.org | |
| Ilastik1.3.2 | https://www.ilastik.org | |
| **Other** | | |
| Leica DM5500 Microscope | Leica Microsystems | |

## Methods and Protocols

### Cell lines and culture condition

For maintenance, wild-type mouse NIH3T3 fibroblasts were cultured in Dulbecco's modified Eagle medium (DMEM, Gibco) complemented with 10% foetal bovine serum (FBS, Sigma) and 1% PSG antibiotics (Gibco). Cells were grown at 37°C, 5% $CO_2$ and 100% humidity and passaged every 2–3 days, until the day of the experiment. Circadian clock entrainment by temperature cycles was performed using a Memmert INCO153 incubator controlled by the Memmert Celsius 10.0 software. We entrained the clock using temperature cycles ranging from 35.5°C to 38.5°C, as previously described (Saini *et al*, 2012) and for at least 10 days prior beginning of the experiment. To maintain synchronisation after passaging, we always applied the splitting procedure at the same temperature cycle position (37°C), when the clock was at its trough. To enhance the synchronisation when splitting, we applied a serum shock by recovering trypsinised cells into DMEM without serum prior inoculating a new dish containing serum-supplemented DMEM (Balsalobre *et al*, 1998).

### Single-molecule RNA fluorescence in situ hybridisation (smRNA FISH)

Cells were plated on 6-well plates containing 18 mm round cover glasses (Fisher Scientific) treated with a solution of 25 μg/ml Fibronectin (Sigma-Aldrich) diluted in 1× PBS for 30 min at room temperature. Wells were seeded with $0.2 \times 10^6$ cells. Cells were cultured in serum-free DMEM (Gibco), supplemented with 1% PSG antibiotics, to prevent cell division. Ten hours later, cells were treated with 100 nM dexamethasone (Sigma-Aldrich) for 30 min. The treatment was followed by medium change. For the experiment where the circadian clock was synchronised using temperature cycles, the same procedure was applied except entrained cells were not treated with dexamethasone. The smRNA FISH protocol was largely adapted from the Stellaris RNA FISH protocol for adherent cells, which describes the method used by Raj *et al* (2008). Stellaris exonic probes coupled with Quasar570 (548/566nm, Red, BMAL1[Red], NR1D1[Red]) or Quasar670 (647/670nm, FarRed, BMAL1[FarRed], CRY1[FarRed]) were hybridised to targeted mRNA. Probe sets are shown in Table EV1. The hybridisation was performed in 50 μl of Stellaris hybridisation buffer complemented with 250 μg/ml Yeast tRNA (Ambion) and 5 mM Ribonucleoside Vanadyl Complex (New England Biolabs). Cells and nuclei were then co-strained with 0.4 μg/ml green HCS CellMask (Invitrogen) and 0.7 μg/ml DAPI (Thermo Fisher), respectively. HCS CellMask and DAPI were diluted in the Stellaris wash buffer. Cover glasses were finally mounted onto microscopy slides in ProLong™ Gold Antifade Mountant (Thermo Fisher). For each time point in both time course experiments (4 and 6 h sampled experiments), cells were plated in triplicate. Each replicate was independently synchronised and prepared for smFISH analysis.

### Imaging

Slides were imaged at the EPFL imaging facility (BIOP) with a Leica DM5500 wide field microscope equipped with a LED Lumencor SOLA lamp, an HCX PL APO 63× Oil objective and an appropriate set of filters (blue, green, orange and far red). We took series of about 40 z-sections with a step of 0.3 μm, depending on the thickness of the cell layer in the field.

### Image processing and data analysis

smRNA FISH microscopy images were processed using a custom analysis pipeline based on open-source software Ilastik1.3.2 and CellProfiler3.1.8 (Kamentsky *et al*, 2011; Berg *et al*, 2019).

1   CellProfiler was first used to generate maximum *Z*-stack projections (all channels) and sum *Z*-stack projections (Red channel only).
2   Secondly, Ilastik pixel classification projects were used to generate probability maps of dots from max *Z*-stack projections. One Ilastik workflow has been used for each (Red) and (FarRed) channels using Random Forest Model trained with a mix of images from independent experiments.
3   Finally, CellProfiler was used to segment dots (dots probability maps), cells (sum Z-projections, Red channel), nucleus (max Z-projection, DAPI channel) and cytoplasms (cell minus nucleus) and related with each other. Data analysis was performed using custom R and Matlab scripts and functions.

### Cosinor regression of smFISH data population mean mRNA levels

To model the smFISH population average mRNA levels (Fig 1B), we performed a least-squares fit using the mean mRNA count of each replicate at all the time points. We used the following function with 2 harmonics to model to mean mRNA level $m_g(t)$ for each gene *g*:

$$m_g(t) = \frac{a_{0,g}}{2} + a_{1,g}\cos\left(\frac{2\pi t}{12} - \phi_{1,g}\right) + a_{2,g}\cos\left(\frac{2\pi t}{24} - \phi_{2,g}\right) \quad (1)$$

### Probabilistic models of smFISH data and parameter inference

To model the smFISH mRNA count distributions from populations of cells, we proposed several candidate models with unique biological interpretations and estimated parameters in a Bayesian paradigm. To infer the parameters, we will use Bayes' rule:

$$p(\theta|y) \propto p(y|\theta)p(\theta).$$

where $p(y|\theta)$ is the likelihood of observing the data given parameters $\theta$ (likelihood), $p(\theta)$ is the estimate of the parameter values before observing evidence (prior probability), and $p(\theta|y)$ is the probability of the parameters $\theta$ after the data $y$ is observed (posterior probability). Note that in the case of normally distributed errors (a Gaussian model), the (log) likelihood reduces to a least-squares model, but this is not applicable here for count data. In our dataset we measure the smFISH count $y_{i,g}$ for cell $i$ and gene $g$, and each cell has an associated time $t_i$ and a measured cell area $A_i$. To calculate the likelihood of the data for a given set of parameters $\theta$, we multiply over a total of $N$ measured cells and all genes

$$p(y|\theta) = \prod_{g}\prod_{i=1}^{N} p(y_{i,g}|t_i, A_i, \theta)$$

where $p(y_{i,g}|t_i,A_i,\theta)$ represents the probability of observing an smFISH count $y_{i,g}$ in a single cell given the time $t_i$, area $A_i$ and parameters $\theta$ for a given model. Once the likelihood and priors are specified for each model, we used the Hamiltonian Monte Carlo sampler provided within the STAN probabilistic programming language (Carpenter *et al*, 2017) to sample model parameters from the posterior distribution using 4 different chains with 1,000 samples each.

### Explicit mathematical models of smFISH counts

For all four models, we use the fact that mRNA counts in a single cell are well described by a negative binomial (NB) distribution, which is an approximation of the telegraph model of bursty transcription that is valid when the mRNA half-life is long in relation to the time spent in the active promoter state (Raj *et al*, 2006), which is typical for mammalian genes (Suter *et al*, 2011; Zoller *et al*, 2015). As we explain now, the NB distributions cover the intrinsic noise, while extrinsic noise sources are modelled with mixtures of such NB distributions, i.e. weighted sums of NB distributions with parameters that can vary from cell-to-cell.

For each model, there is a burst size parameter $b_{i,g}$ and bust frequency parameter $f_{i,g}$, which depend on the model considered (Table 1) and which control the shape of the mRNA distribution for each gene $g$. Since 3T3 cells are tetraploid, and, again assuming that the bursts are short, the inferred burst frequency for tetraploid cells will be approximately four times that of a single allele. The probability of observing an smFISH count $y_{i,g}$ for gene $g$ and for cell $i$ follows a negative binomial distribution:

$$p\left(y_{i,g}|b_{i,g},f_{i,g}\right) = Neg - bin\left(y_{i,g}|b_{i,g},f_{i,g}\right)$$
$$= \frac{\Gamma(f_{i,g}+y_{i,g})}{\Gamma(y_{i,g}+1)\Gamma(f_{i,g})}\left(\frac{1}{1+b_{i,g}}\right)^{f_{i,g}}\left(\frac{b_{i,g}}{1+b_{i,g}}\right)^{y_{i,g}}$$

The mean of the NB distribution is given by $b_{i,g}f_{i,g}$, while the variance is given by $b_{i,g}f_{i,g} + b_{i,g}^2 f_{i,g}$.

### Model M1—oscillatory burst frequency and fixed burst size

In model M1, the burst size ($b_g$) is a constant while the burst frequency ($f_g$) varies with circadian time, which is based on previous live-cell imaging experiments using the same cell line showing that burst frequency of clock genes is modulated by the circadian clock while the

burst size remains constant (Nicolas *et al*, 2018). For all models, the burst frequency ($f_g$) uses the same function $m_g(t)$ (Equation 1) that was fitted to the mean mRNA level but rescaled with a parameter $\gamma_g$, which ensures that the phase of oscillations is the same for all models. We used weakly informative priors on the parameters $b_g, \gamma_g \sim N(0,100)$. The complete probabilistic model for M1 is therefore.

$$b_g \sim Normal(0,100)$$
$$\gamma_g \sim Normal(0,100)$$
$$b_{i,g} = b_g$$
$$f_{i,g} = \gamma_g m_g(t_i)$$
$$y_{i,g} \sim Neg - bin(b_{i,g}, f_{i,g})(M1)$$

### Model M2—cell size-dependent burst size

Model (M2) incorporates cell size, where we modified the dependence between burst size and cell area with an exponent $\beta_g$. This assumption is based on previous results in mammalian cells showing that burst size scales with cell volume (Padovan-Merhar *et al*, 2015), and the use of an exponent $\beta_g$ is to account for the fact that we measure cell area instead of cell volume. Consequently, the reformulated model is as follows:

$$b_g \sim Normal(0,100)$$
$$\gamma_g \sim Normal(0,100)$$
$$\beta_g \sim Normal(0,100).$$
$$b_{i,g} = b_g A_i^{\beta_g}$$
$$f_{i,g} = \gamma_g m_g(t_i)$$
$$y_{i,g} \sim Neg - bin(b_{i,g}, f_{i,g})(M2)$$

where $A_i$ represents the cell area for cell $i$, and we again used a weakly informative prior $\beta_g \sim N(0,100)$.

### Model M3—phase noise from incomplete synchronisation

For model M3, we incorporate imperfect synchronisation by allowing each cell to have an individual phase $\varphi_i$, and the cell-specific phases are distributed around a population average. We used a von Mises prior $\varphi_i$ for the distribution of cell phases.

$$VonMises(\varphi_i|k) = \frac{e^{\kappa\cos(\varphi_i)}}{2\pi I_0(\kappa)}$$

where $I_0(\kappa)$ is the modified Bessel function of order 0. We used a value of $\kappa = 2$, which approximately matches the phase spread observed in live microscopy for mammalian fibroblasts (Pulivarthy *et al*, 2007). To simplify the analysis, we fixed parameters previously contained in model M2 to their posterior mean values, and in M3, we keep the waveform shape and phase for the bursting frequency fixed (from M2), but allow the burst frequency amplitude to vary. The amplitude of the oscillatory function is rescaled with the parameter $\lambda_g$, and the full model is thus given by

$$\lambda_g \sim U(0,2)$$

$$\varphi_i \sim VonMises(\varphi_i | \kappa = 2)$$

$$b_{i,g} = b_g A_i^\beta$$

$$f_{i,g} = \gamma_g \left( \frac{a_{0,g}}{2} + \lambda_g a_{1,g} \cos\left( \frac{2\pi t_i}{24} - \phi_{1,g} - \varphi_i \right) + \lambda_g a_{2,g} \cos\left( \frac{2\pi t_i}{12} - \phi_{2,g} - \varphi_i \right) \right)$$

$$y_{i,g} \sim Neg-bin(b_{i,g}, f_{i,g})(M3)$$

### Model M4—cell-specific burst size

In model M4, we included an additional cell-specific random variable $\epsilon_{i,g}$ to modify the burst size in each cell $i$ for each gene $g$. We assumed that the $\epsilon_{i,g}$ is bivariate log-normally distributed between pairs of genes $g = 1$ and 2. The log-normal distribution is parameterised with $\mu_g$, $\sigma_g$ and $\rho$, which represent the mean, variance and correlation of $\epsilon_{i,g}$ between genes $g_1$ and $g_2$ (in log space). Practically, this means that when bursts are large for one gene, they can be large (correlated) or small (anticorrelated) for another gene. Biologically, this correlation could be generated when the transcription rate of one gene is affected by the activity of another gene (i.e. feedback). Model M4 is therefore specified as.

$$\mu_g \sim Normal(0,100)$$

$$\sigma_g \sim Normal(0,100)$$

$$\rho \sim LKJ(4.0)$$

$$\log \begin{pmatrix} \varepsilon_{i,g_1} \\ \varepsilon_{i,g_2} \end{pmatrix} \sim N \left( \begin{bmatrix} \mu_{g_1} \\ \mu_{g_2} \end{bmatrix}, \begin{bmatrix} \sigma_{g_1}^2 & \rho \sigma_{g_1} \sigma_{g_2} \\ \rho \sigma_{g_1} \sigma_{g_2} & \sigma_{g_2}^2 \end{bmatrix} \right)$$

$$b_{i,g} = \varepsilon_{i,g} A_i^{\beta_g}$$

$$f_{i,g} = \gamma_g m_g(t_i)$$

$$y_{i,g} \sim Neg-bin(b_{i,g}, f_{i,g})(M4)$$

We used an LKJ prior with shape parameter 4 to regularise $\rho$. The correlation coefficient $\rho$ is inferred between *Bmal1-Cry1* and *Nr1d1-Cry1*, but it is not directly inferred between *Nr1d1* and *Bmal1*. As such, for the 3-dimensional simulations we assume the $\epsilon_{i,g}$ parameters are conditionally independent between these two genes (i.e. the entry in the precision matrix for *Nr1d1-Bmal1* is zero).

### Model selection

We use two different methods to estimate the predictive accuracy of all of our models: adjusted within-sample predictive accuracy and cross-validation. The starting point for our first approach is given by the *elpd* (expected log pointwise predictive density for a new dataset). This can in theory be estimated using leave-one-out cross-validation, where one cell is removed from the dataset.

$$elpd_{\text{leave-one-out}} = \sum_{i=1}^N \log p(y_i | y_{-i}) = \sum_{i=1}^N \log \int p(y_i | \theta) p(\theta | y_{-i}) d\theta$$

which uses the leave-one-out predictive density given the data without cell $i$. However, this would involve retraining the model

for each cell, which is not feasible with thousands of cells. As shown in (Gelfand *et al*, 1992), the log predictive density for the held-out cell $y_i$ can be estimated from the posterior samples of the full dataset using importance ratios

$$p(y_i | y_{-i}) \approx \frac{1}{\frac{1}{S} \sum_{s=1}^S \frac{1}{p(y_i | \theta^s)}}$$

We estimate the $elpd_{\text{leave-one-out}}$ with Pareto smoothing of the importance weights (Pareto-smoothed important sampling, PSIS-LOO) using the parameter samples for each model together with the PSIS-LOO python function provided with (Vehtari *et al*, 2017).

We also used cross-validation directly by performing parameter inference for each model on a training set and then calculating the log-likelihood of a test set. We partitioned the data by removing one gene from each replicate slide $k$ from a total of $K$ slides. We performed posterior parameter sampling on the training data, and then, we used the posterior mean $\hat{\theta}_{-k} = \int \theta p(\theta | y_{-k}) d\theta$ to calculate the likelihood of observing the test replicate data $\log p(y_k | \hat{\theta}_{-k})$. The expected log pointwise predictive density for a new slide is then given as.

$$elpd_{\text{leave-replicate-out}} = \sum_{i=1}^N \log p(y_k | \hat{\theta}_{-k})$$

Model M4 contains latent variables $\epsilon_k$ that are not known for the new gene on the test slide $k$. We therefore integrate over latent variables $\epsilon_k$ using $S$ Monte Carlo simulations.

$$elpd_{\text{leave-replicate-out}} = \sum_{k=1}^K \log \int p(y_k, \epsilon_k | \hat{\theta}_{-k}) d\epsilon_k = \sum_{k=1}^K \log \frac{1}{S} \sum_{s=1}^S p(y_k | \epsilon_k^s, \hat{\theta}_{-k})$$

### Variance decomposition

The goal of variance decomposition is to determine how intrinsic noise and fluctuations in extrinsic parameters contribute to the variance in smFISH counts. We follow the approach shown in (Bowsher & Swain, 2012) that generalises the Law of Total Variance to any number of variables. In this framework, intrinsic variance will be defined as the average variance of gene expression in cells with exactly the same intracellular conditions, and extrinsic variability is the variance generated by cellular parameters and additional stochastic variables in the cell. For the selected model M4, extrinsic noise refers to time $t$, cellular area $A$ and the burst size noise term $\epsilon$, whereas intrinsic noise refers to the variance of the negative binomial distribution averaged over all other noise sources. Practically, the variance is decomposed by successively conditioning on groups of extrinsic parameters. For mRNA counts $Y$ and extrinsic parameters $X_1$, $X_2$ and $X_3$ the decomposition is as follows:

$$V[Y] = E[V[Y|X_1,X_2,X_3]] + E[V[E[Y|X_1,X_2,X_3]|X_1,X_2]]$$

$$+ E[V[E[Y|X_1,X_2]|X_1]] + V[E[Y|X_1]],$$

where the first term represents the variance from intrinsic transcriptional noise, and the remaining terms describe the contribution to variance from extrinsic variables. The expectation operator $E$ used with conditioning variables (i.e. $E[Y|X_i]$) denotes averaging over all random variables except those given in the conditioning. In our case, the extrinsic conditioning variables $X_i$ correspond to time $t$, cellular area $A$

and the burst size noise term $\epsilon$. There is no unique mathematical way to decompose the variance with multiple extrinsic variables, and hence, the contribution to variance of each of our extrinsic variables depends on whether the extrinsic conditioning variables $X_i$ is assigned to $X_1$, $X_2$ or $X_3$. We therefore average over all possible combinations, but practically the difference in variance calculated under different orderings, e.g. $E[V[E[Y|X_1,X_2,X_3]|X_1,X_2]]$ compared to $V[E[Y|X_3]]$ is usually only a small percentage.

The intrinsic transcriptional noise contains the term $V[Y|X_1,X_2,X_3]$, which is the variance of the negative binomial distribution for given values of extrinsic parameters. This term can be further decomposed into a Poisson and a promoter noise component:

$$V[Y|X_1,X_2,X_3] = E[Y|X_1,X_2,X_3] + \alpha E[Y|X_1,X_2,X_3]^2,$$

where $\alpha$ is the dispersion parameter of the negative binomial distribution. Under our model, the expectation $E[Y|X_1,X_2,X_3]$ is equal to $bf$ and the dispersion parameter $\alpha$ is equal to $1/f$. All terms were computed via simulation of the inferred model, where we used the posterior mean parameter values of the preferred model M4.

### A simplified stochastic model of the core clock circadian network

In this section, we describe the simplified stochastic model of the *Nr1d1/Cry1/Bmal1* reaction network that is used in Appendix Fig S13. The basis of the model is the telegraph model of bursty gene expression (Peccoud & Ycart, 1995). For each gene, the promoter can be in an active state (denoted $g_{on}$) or an inactive state ($g_{off}$). Production of mRNA molecules (denoted $M$), i.e. transcription, only occurs when the promoter is in an active state, and the mRNAs are then translated into protein molecules ($P$).

To model the transcription factor network, the protein molecules of each gene can activate or repress the transcription rate of other genes or itself (i.e. autoregulation). To use a realistic network topology, we adapted the mathematical model proposed in (Relógio *et al*, 2011) (network topology shown in Appendix Fig S13A).

To keep the model close to the preferred model M4, we used a 24-h oscillating function $k_{on}(t)$ to control the promoter activation reaction $g_{off} \xrightarrow{k_{on}(t)} g_{on}$ (and hence the burst frequency is oscillating). We must therefore use a simulation framework that accounts for time-varying hazard functions, and to do this, we use a thinning approach (Lewis & Shedler, 1979; Voliotis *et al*, 2016). The algorithm used for simulation is shown in Algorithm 1. In essence, a "thinning" reaction is added with hazard function $w(t)$ such that $k_{on}(t)+w(t)$ is a fixed

$k_{on}(t)$. We keep track of the number of thinning events by introducing an additional species $R$ for each gene. The total state of the network can then be described with the vector:

$$X = [g_{off,Cry1}, g_{on,Cry1}, M_{Cry1}, P_{Cry1}, R_{Cry1}, \dots$$
$$g_{off,Nr1d1}, g_{on,Nr1d1}, M_{Nr1d1}, P_{Nr1d1}, R_{Nr1d1}, \dots$$
$$g_{off,Bmal1}, g_{on,Bmal1}, M_{Bmal1}, P_{Bmal1}, R_{Bmal1}]^T$$

For each gene $i$ in $\{Nr1d1,Cry1,Bmal1\}$, the model is then described with the following set of reactions:

$$g_{off,i} \xrightarrow{k_{on,i}(t)} g_{on,i} \quad \text{Promoter activation}$$
$$g_{off,i} \xrightarrow{w_i(t)} g_{off,i} + R_i \quad \text{Thinning reaction}$$
$$g_{on,i} \xrightarrow{k_{off,i}} g_{off,i} \quad \text{Promoter deactivation}$$
$$g_{on,i} \xrightarrow{h_i(X)} g_{on,i} + M_i \quad \text{Transcription}$$
$$M_i \xrightarrow{\alpha P} M_i + P_i \quad \text{Translation}$$
$$M_i \xrightarrow{\delta} \phi \quad \text{mRNA degradation}$$
$$P_i \xrightarrow{\delta} \phi \quad \text{Protein degradation}$$

The gene–gene interactions are encoded through the regulation of transcription rate by the protein levels of other genes in the network ($h_i(X)$). To simplify the model, we used an activatory or inhibitory Hill function to represent each of the feedback loops used in Relógio *et al* (2011) (network topology shown in Appendix Fig S13A).

$$h_{Cry1}(X) = \alpha_{M,Cry1} \left( \frac{1}{1+\left(\frac{P_0}{P_{Bmal1}}\right)^n} \right) \left( \frac{1}{1+\left(\frac{P_{Nr1d1}}{P_0}\right)^n} \right) \left( \frac{1}{1+\left(\frac{P_{Cry1}}{P_0}\right)^n} \right).$$

$$h_{Nr1d1}(X) = \alpha_{M,Nr1d1} \left( \frac{1}{1+\left(\frac{P_0}{P_{Bmal1}}\right)^n} \right) \left( \frac{1}{1+\left(\frac{5P_{Cry1}}{P_0}\right)^n} \right).$$

$$h_{Bmal1}(X) = \alpha_{M,Bmal1} \left( \frac{1}{1+\left(\frac{P_{Nr1d1}}{P_0}\right)^n} \right).$$

We use the oscillatory burst frequency inferred from the data to set the promoter activation rate. The burst frequency inferred for each model $f_i = \gamma_i \left( \frac{a_{0,i}}{2} + a_{1,i} \cos\left(\frac{2\pi t}{24} - \phi_{1,i}\right) + a_{2,i} \cos\left(\frac{2\pi t}{12} - \phi_{2,i}\right) \right)$ is rescaled by the mRNA half-life parameter $\delta$.

$$k_{on,Cry1}(t) = \delta \gamma_{Cry1} \left( \frac{\alpha_{0,Cry1}}{2} + \alpha_{1,Cry1} \cos\left(\frac{2\pi t}{24} - \phi_{1,Cry1}\right) + \alpha_{2,Cry1} \cos\left(\frac{2\pi t}{12} - \phi_{2,Cry1}\right) \right)$$

$$k_{on,Nr1d1}(t) = \delta \gamma_{Nr1d1} \left( \frac{\alpha_{0,Nr1d1}}{2} + \alpha_{1,Nr1d1} \cos\left(\frac{2\pi t}{24} - \phi_{1,Nr1d1}\right) + \alpha_{2,Nr1d1} \cos\left(\frac{2\pi t}{12} - \phi_{2,Nr1d1}\right) \right)$$

$$k_{on,Bmal1}(t) = \delta \gamma_{Bmal1} \left( \frac{\alpha_{0,Bmal1}}{2} + \alpha_{1,Bmal1} \cos\left(\frac{2\pi t}{24} - \phi_{1,Bmal1}\right) + \alpha_{2,Bmal1} \cos\left(\frac{2\pi t}{12} - \phi_{2,Bmal1}\right) \right)$$

upper bound $U_{k_{on}}$. The sum of the two reactions is then a homogenous Poisson process with rate $U_{k_{on}}$. By starting with a homogenous Poisson process with rate $U_{k_{on}}$ and keeping each event with probability $k_{on}(t)/U_{k_{on}}$, the resulting process will be inhomogenous with rate

All model parameters are shown in Appendix Table S1. Stochastic simulations are then performed with Algorithm 1.

Algorithm 1: Stochastic simulations of the network with time-varying parameters.

| Initialise time $t \leftarrow 0$ and network state $X \leftarrow X_0$; |
| :--- |
| **while** $t \leq T$ **do** |
| Set $a_0 \leftarrow \sum_{j=1}^{M} a_j[X2Ct]$, where $a_j[X,t]$ are the reaction propensities $a_j$ at time $t$; |
| Draw exponentially distributed number $\tau \sim \exp(1/a_0)$; |
| Update time $t \leftarrow t + \tau$; |
| Update all propensities $a_j[X,t]$; |
| Set $a_0 \leftarrow \sum_{j=1}^{M} a_j[X2Ct]$; |
| Draw uniformly distributed random number $u \sim U(0,1)$; |
| Choose reaction associated with the smallest positive integer $j$ satisfying: $\sum_{i=1}^{j} a_j[X2Ct] \geq a_0 u$; |
| **end** |

## Data availability

The datasets and computer code produced in this study are available in the following databases: Modelling computer scripts: GitHub (https://github.com/naef-lab/CircadianSMFISH).

**Expanded View** for this article is available online.

## Acknowledgements

We thank Clémence Hurni and the BIOP imaging facility at the EPFL for support in setting up the multichannel smFISH imaging and members of the Naef-lab for valuable feedback on the manuscript. Research in the Naef lab is supported by a Swiss National Science Foundation Grant number 310030_173079 and the EPFL.

## Author contributions

NEP, AH and FN conceptualised the study. NEP, AH, JY and FN contributed to the methodology. AH, ED and DN performed investigation. NEP and AH performed formal analysis. ED and AH curated the data. NEP, ED, AH and JY contributed to the software. NEP, AH and FN wrote the original draft. NEP, AH, JY, ED, DN and FN reviewed and edited the manuscript. FN provided resources and supervision.

## Conflict of interest

The authors declare that they have no conflict of interest.

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
