## [Review Process File · Molecular Systems Biology]

The circadian oscillator analysed at the single-transcript level

Nicholas Phillips, Alice Hugues, Jake Yeung, Eric Durandau, Damien Nicolas, and Felix Naef
DOI: [10.15252/msb.202010135](https://doi.org/10.15252/msb.202010135)

Corresponding author(s): Felix Naef (felix.naef@epfl.ch)

Review Timeline:

Transfer from Review Commons:	20th Nov 20
Editorial Decision:	21st Dec 20
Revision Received:	5th Jan 21
Accepted:	19th Jan 21

Editor: Maria Polychronidou

Transaction Report: This manuscript was transferred to Molecular Systems Biology following peer review at Review Commons.

Review
COMMONS

Reviewer #1 (Evidence, reproducibility and clarity (Required)):

The authors generated and analyzed a great amount of single-cell RNA FISH data over time on circadian genes (*Nr1d1*, *Cry1*, *Bmal1*), and performed model selection/fitting to explain the observed mRNA distributions. They decomposed the mRNA variability into distinct sources, and showed that intrinsic noise (transcription burst) dominates the variance. Therefore, looking at transcript counts may not be feasible to estimate single-cell circadian phase. However, the study is quite descriptive and ends up being a bit dissatisfying, so if the authors could improve this aspect by perhaps analyzing a mechanism on cell-specific burst size ($F5$), gene-specific dependence on cell size (β), or the positive/negative gene-pair correlations (ρ), it would help quite a bit in this regard. The model selection/fitting itself was not really sufficient to compensate for this, as it stands.

We thank the reviewer for appreciating the new smFISH data, the analyses performed, and the consequences regarding phase inference from single cell snapshots.

The reviewer suggests “perhaps analyzing a mechanism on cell-specific burst size ($F5$), gene-specific dependence on cell size (β), or the positive/negative gene-pair correlations (ρ)”, and we have thus added a new Results paragraph (lines 281-316) and two new Supp Figures 13 and 14 to directly address this point.

Specifically, we have added a dynamic, stochastic model of the circadian clock in order to add mechanistic insight into the parameters of the preferred model M4. Concerning ρ , in the initial manuscript we suggested that the correlations of cell-specific burst sizes (described by the parameter ρ) in the preferred model M4 could result from the underlying network topology. To substantiate this claim, we have now added an analysis of a stochastic model of the clock that includes gene-gene interaction amongst the core-clock genes. The core-clock network involves variables (such as protein levels), parameters (such as mRNA/protein half-lives) and additional genes (such as *Clock*) that are not directly measurable in our experiments; and thus offering a detailed mechanistic mathematical model for our data is therefore not realistic. We therefore developed a simplified mathematical model for the three measured genes to explore the underlying mechanisms that could control the parameter ρ , as the referee suggests. As a starting point, we used the circadian clock gene network topology for *Nr1d1*, *Cry1* and *Bmal1* as modelled in Relógio et al. (Relógio et al., 2011) (see new Supplementary Material). To keep the model close to the inference framework, we used oscillatory functions for the burst frequency while the transcription rate (and hence the burst size) for each gene is affected by the protein levels of the other genes in the network. Using stochastic simulations we show that, for particular configurations of feedback where the negative repression of *Nr1d1* by CRY1 is high, the network can generate positive mRNA correlation between *Bmal1/Cry1* mRNA and negative correlation between *Nr1d1/Cry1* mRNA, as observed in our data (Figure 2C). Furthermore, using the same inference framework as for our data on the simulated mRNA distributions, the obtained ρ is positive for *Bmal1/Cry1* and negative for *Nr1d1/Cry1*, which was also found for our data (Figure 3C). Even though the model is clearly a simplified representation of the clock, these simulations give credence to the scenario that the ρ parameter obtained from the data is a signature of the underlying network topology.

While the emphasis of the paper is certainly on parameter inference of the single-cell RNA FISH data, we believe the addition of this dynamic model provides more mechanistic insight into the results of the model fitting and hence significantly more depth to the article.

****Specific comments:****

1. It is hard to distinguish the RNA FISH signals (Figure 1A, 2B). It is probably technically challenging as the mRNAs are of low abundance. I think it may help if they adjust the contrast for the cytoplasm stain or just delineate the cell boundaries.

Thank you for pointing this out, and we agree that our rendering of the FISH images was not optimal and have now significantly improved it (see new Figure 1A and 2B). Considering the other reviewers' comments related to the images, we have now 1) added the cell contours as requested; 2) use red/green for the smFISH signal in the pairs of genes; 3) we have improved the contrast to make it easier to distinguish the RNA FISH signals.

2. In Figure 2C, the authors showed gene-pair correlations with cells of all sizes. Could the authors do a size-dependent extrinsic-noise filtering (Padovan-Merhar, Dev. Cell, 2015; Hansen et al., 2018, Cell Systems) to better dissect the correlations?

We used negative binomial distributions to directly model the number of mRNA in the cells, which is a natural choice given that the raw smFISH are integer counts. The model incorporates cell size dependencies in a unified framework, which predicts the joint distribution of raw counts, which is why we showed raw counts in the main figure. That being said, as the referee suggests, it can be useful for exploratory purposes to see the relationship between the measured genes while regressing out the contribution of cell area, and we have now added this analysis as Supp Figure 9. On line 156-161 we write: "To also estimate the correlation between genes while accounting for cell area, we regressed out the area for each gene and recalculated the correlation coefficients [37,38]. Since all genes are positively correlated with area (Fig. 2A), this processing shifted the correlations for both pairs of genes. Specifically, the correlation coefficients for the area-filtered mRNA counts decreased but remained positive for *Bmal1/Cry1* and became more negative for *Nr1d1/Cry1* (Supp Figure 9)."

3. For fitting model M3, as the authors pointed out, there are many local minima. Is the fitting score truly sufficient to eliminate the possibility for partial synchrony especially considering that the authors didn't show how effective the Dex treatment was to synchronize the circadian phase?

Thank you for this comment. In fact, we didn't mean to fully eliminate the possibility of imperfect synchronization, but have tried our best to address it both experimentally and with modeling.

Experimentally, in addition to the Dex treatment, we also compared with a condition in which we entrained the cells using temperature cycles, which is a standard in the field to achieve the best synchronization. We obtained a fold change of 2.1, which was in the range of previous studies (Saini, et al, 2012) and was slightly higher than with Dex synchronisation (1.6). Given that the improvement was not high and that it was important for us to study the

system under free-running conditions and not in an entrained state (i.e. phase locking, which distorts the free dynamics and noise characteristics of the oscillator), we used the Dex protocol.

Model 3 was used as a computational approach to correct for the individual phases. In addition to the difficult optimisation landscape, the challenge with model M3 also resides in the difficulty of estimating an individual phase for each cell, as the two mRNA counts measured in each cell do not contain sufficient phase information. This could potentially be resolved by either measuring more genes simultaneously, but is, however, beyond the scope of the present manuscript. We have added discussion on this to the text on lines 244-248:

“Thus, it was apparently difficult to use model M3 to correct the individual phase for each cell, likely due to the fact that the two mRNA counts measured in each cell do not contain sufficient phase information, and that the global optimisation problem contains many local minima. This could potentially be improved by measuring more genes simultaneously.”

We have also added a new Results section (lines 305-316) and Supp Figure 14 to show that imperfect synchrony alone cannot explain the correlation structure observed in our data. Indeed, if two genes have a similarly phased oscillation, the expression of the two genes will be positively correlated (as shown in the new Supp Figure 14). Similarly, when the oscillations are in anti-phase, negative correlations will be found. Given that *Nr1d1* and *Cry1* are closer in phase than *Bmal1* and *Cry1*, one would expect that the correlation between *Nr1d1* and *Cry1* (once accounting for area) would be more positive than for *Bmal1* and *Cry1*, which was not found in the data (area-corrected correlations shown in Supp Figure 9). It therefore seems unlikely that the observed correlations could be caused by imperfect synchrony alone. Together with our simulations of the gene network (described above), we therefore argue that gene-gene interactions are a more plausible mechanistic explanation of the correlations observed in our measured bivariate mRNA distributions.

4.Regarding model M4, the authors added a cell-specific noise term without specifying the contributing factors. Typically adding degrees of freedom should improve fitting and make it easier for a model to fit, why not in this case? Can the authors provide some explanations/mechanisms.

We believe there has been a misunderstanding regarding model M4. By adding parameters, model M4 is indeed easier to fit. There is even a problem of overfitting whereby the burst frequency becomes unrealistically high and the model effectively fits a Poisson distribution to each individual cell. To avoid this, we lock the burst frequency values to the posterior mean values from model M2. After describing model M4, we write (lines 260-265):

“When all parameters are free, we noticed that the burst frequency can become unrealistically high due to a tendency to overfit to individual cells, and we therefore locked the burst frequency to the posterior mean values from model M2. The PSIS-LOO scores overall favoured model M4 (Fig. 3B), and the predicted joint probability density shows good similarity to the observed data (Fig. 3D) (all time points shown in Supp figure 11).”

Regarding the above comment in the reviewer's summary on contributing factors of model M4 we added a simple dynamical model that attempts to explain at least one possible mechanism of generating correlations in cell-specific bursting parameters (see above).

5. The authors should include the number (range) of cells analyzed in the figure legends.

We have now added the number of cells used at each time point to the legend of Figure 1D. To respond to Reviewer #2 we have also added details on the number of smFISH replicates used at each time point. The number of cells for each replicate is shown in Supp Figures 2-5.

Reviewer #1 (Significance (Required)):

Overall, we felt conflicted about the manuscript. On one hand, the authors generated and analyzed a great amount of single-cell RNA FISH data over time on circadian genes. On the other hand, the manuscript was a bit dissatisfying/descriptive. If the authors could provide and analyze some sort of mechanisms on cell-specific burst size ($F5$), gene-specific dependence on cell size (β), or the positive/negative gene-pair correlations (ρ) it should help improve the manuscript.

We thank the review for the suggestion to expand on the mechanistic interpretation, which we have followed. In addition, we would like to emphasise that a similar smFISH analysis of the core circadian oscillator has never been done, and we believe our data represents a significant contribution to the field. Moreover, our quite generic probabilistic inference framework for smFISH using mixture models to describe intrinsic (transcriptional bursting) and extrinsic fluctuations is also novel and the code provided (written using the Stan probabilistic programming language) might find a wide applicability.

Concerning the mechanistic description, as described above, we added a stochastic, dynamic model of gene expression and propose that gene-gene interactions within the core-clock network topology represent a plausible mechanism for generating correlated burst parameters between genes, which are a feature of the preferred model M4 found during inference. We additionally added an explanatory figure to argue that, given the phase relationship between genes, imperfect synchronisation alone cannot explain the observed correlations that we observe between the pairs of genes. Together, this analysis provides more mechanistic insight into the underlying factors controlling the gene-gene relationships in our measured bivariate mRNA distributions.

Referees cross-commenting

I agree with Reviewer #3 regarding expanding the discussion to include the Shah & Tyagi and Raj et al citations on buffering. However caution should be exercised regarding ref 26 as it is quite controversial and subsequent analyses came to different conclusions (PMID: 30359620 and 30243562). The general consensus is that nuclear buffering of transcript noise (proposed in ref 26) is not a general phenomenon (ref 27 is specific to the calcium response pathway). In fact, the presence and evolution of specific pathways to buffer transcriptional noise, such as protein-protein mechanisms (Shah & Tyagi) or extended half-

life proteins (Raj et al. and others), argues that transcript fluctuations are not probably buffered in general.

Following the suggestion of Reviewer #3, we have expanded the Discussion to include the references cited (Shah & Tyagi, Raj and others).

Previous work from our lab is also nuancing the conclusions from references 26 and 27. Specifically, buffering effects are expected to be highly gene-specific (3'UTR), and in fact we have not seen those with our unstable construct during live-cell imaging (Suter et al., 2011; Zoller et al., 2015). We have also added text in order to explicitly state that subsequent papers have nuanced the general claims in references 26 and 27. In the text we write (lines 335-342):

“One explanation for the low intrinsic fluctuation in these studies is that transcriptional fluctuations are filtered by nuclear retention, though other reports suggest that Fano factors (variance/mean, a measure of overdispersion compared to the Poisson distribution) can be even larger in the cytoplasm than in the nucleus [38]. In the cells used here, the strong signature of transcriptional bursting and high intrinsic noise is consistent with live imaging of a *Bmal1* transcriptional reporter in the same cell line under similar growth conditions, where intrinsic noise was estimated to be 4-times larger than extrinsic noise [23].”.

Reviewer #2 (Evidence, reproducibility and clarity (Required)):

*****Summary:*****

The authors study experimentally and computationally the dynamic transcription of circadian clock genes over time in individual cells with single molecule RNA-FISH with the aim to understand how different noise sources contribute to single cell transcription variability and basic functions of circadian clocks. The authors integrate experiments with computational modeling to understand biology.

*****Major comments:*****

This study has some major limitations that need to be addressed to test the model usefulness, to understand noise sources and to gain biological insights into circadian clocks.

We thank this reviewer for the constructive feedback which enabled us to significantly strengthen the revised manuscript.

The limitations are on the experiments, the computational implementation of the modeling and the integration of experiments with models.

Although the experimental datasets contain several hundred cells per time point for multiple time points, only a single replica experiment is presented. From the presented data it is not clear how reproducible these temporal patterns are and if indeed differences between timepoints can be resolved if multiple biological replica experiments have been analyzed. To

address this point at least three biological experiments needs to be presented and analyzed for each of the genes. Plotting the SEM on the means in figure 1B is misleading because several hundred cells have been measured which automatically makes the error small. The SEM just describes how well we can determine the mean from a distribution. Instead a mean and std from the biological replicas need to be plotted to show how experimental variability in experiments is resulting in the described expression pattern. This is similar to RNA-seq data or RT-PCR from multiple replica.

We certainly agree that demonstrating reproducibility is important. Note that our smFISH data is from three independent cell culture dishes and microscopy slides, which included independent cell synchronization. This was described in the Methods but we agree that the data presentation was not showing the individual replicas, which we have now added. In Figure 1B, we now show the mean of each replicate for each time point. While the reviewer suggested displaying the mean and standard deviation across replicates, we show all data points at each time point to make it even more transparent. The mRNA distribution of each replicate is also shown in Supp Figures 2-5, together with individual quantification of mean, coefficient of variation and number of cells.

In addition, to further demonstrate the reproducibility of the temporal patterns we have performed an additional independent experiment on four time points. This experiment shows that the oscillatory patterns for *Nr1d1* and *Cry1* are clearly significant and reproducible (new Supp Figure 7). The combination of the replicates shown for the main experiment (Supp Figures 2-5) and the new replicate experiment (Supp Figure 7) shows that the oscillatory temporal patterns for the mean mRNA levels are robust and reproducible, and in fact similar as those found in bulk analyses (Ukai-Tadenuma et al., 2011; Hughes et al., 2009), which is expected.

It is also not clear how good the cell segmentation works and how does cell segmentation influence the analysis. In figure 1A show the segmentation of the cell boundary together with the membrane stain.

Thanks to this and other reviewers' comments, we have now significantly improved the presentation of the FISH images. We have now 1) added the cell contours as requested; 2) used red/green for the smFISH signal in the pairs of genes; 3) we have improved the contrast to make it easier to distinguish the RNA FISH signals.

We have also added Supp Figure 1 to show that the cell segmentation we used is reliable. In fact, as we had described, we used the sum Z-stack projections of the red channel (Wu et al., 2018), which we found provides the most accurate cell segmentation. We now show in Supp Figure 1 that the obtained segmentation shows convincing agreement with the cell autofluorescence .

The authors use the RNA mean and RNA-FISH distributions and combine this data to build and compare different models. How do you know that the given data fulfils the central limit so that a model describing the mean is an adequate approach? To test this point, the authors should show through subsampling from the data and the model that indeed their data sets have enough cells to fulfil the central limit theorem.

This comment reflects a misunderstanding of our approach, which we now try to better explain. In our inference framework we use a negative binomial (NB) distribution (and mixtures of NBs) to model the full distribution of mRNA counts, and our approach is therefore not based exclusively on the mean of the distribution. The estimation of model parameters and comparison of models is performed using the PSIS-LOO optimisation procedure (see below). The mixture model of NB binomials makes a few assumptions which we had clearly stated. In fact it captures both bursty transcription (in the limit of short bursts as is biologically plausible, which yields the NB distribution), and cell-to-cell variability (extrinsic noise) captured by the mixture. The suitability of the NB to model bursty transcription is established (Raj et al., 2006), and it is parameterized by a mean and a dispersion coefficient, such that the CV of the distribution is the inverse of the burst frequency (Zoller et al., 2015). Therefore the mean is indeed an important parameter of the model, but we do not see the relationship with the CLT. The used probabilistic inference (PSIS-LOO: Pareto-Smoothed Importance Sampling Leave-One-Out, Vehtari et al. 2017, see below) is established and state-of-the-art for selecting models of the appropriate complexity and we are not aware of a similar previous quantitative model for smFISH analysis.

We have now added significantly more explanations both on the general approach as well as the methodological details in a fully-revised Methods section to avoid further misunderstanding.

A strength of the manuscript is that several competing and biologically meaningful models have been generated. However, the manuscript lacks rigor in terms of how fitting and model selection is performed. It is not clear how good the models fit the data. To address this point, the authors should visually compare the model fits to the data and plot their fit errors as a function of model complexity.

We fully agree that comparing different models using a model selection approach is a powerful methodology, in fact it is arguably the most systematic way to approach modeling problems in quantitative biology. Model selection is an active research area and there have been significant developments recently. Here, we used a state-of-the-art and established Bayesian approach (PSIS-LOO: Pareto-Smoothed Importance Sampling Leave-One-Out, Vehtari et al. 2017), which is certainly rigorous and more objective than visual comparison. The PSIS-LOO is conceptually similar to other approaches of model performance such as AIC or WAIC, and the entire field of model selection aims at establishing rigorous methods to assess the tradeoff between fit errors and model complexity. In PSIS-LOO, this is done by using pareto-smoothed importance sampling to estimate the expected log pointwise predictive density for a new dataset using leave-one-out cross-validation. The PSIS-LOO is the currently recommended metric for measuring model performance in Bayesian analysis (Vehtari et al., 2017) and is considered superior to other approaches such as computations of Bayes factors since it is less sensitive to model priors (Gelman et al. 2013). The performance of the models as measured with PSIS-LOO is shown in Figure 3B. As already mentioned, we have added further details as to how the fitting and model selection is performed in a revised Methods section. We agree that visual comparison is useful to gain intuition and this is why we showed the bivariate distributions in Figure 3D and in Supp Figure 11.

Regarding the comment on “fit error”, note also that we probabilistically model the full mRNA distribution for each gene. In each cell, there is a likelihood score that measures the likelihood of observing the measured mRNA count given the modelled probability distribution. As our approach is based on this likelihood, the notion of “fitting error” needs to be replaced by the log likelihood (‘fitting error’ is mathematically equivalent to a log-likelihood when the noise model is Gaussian, which is not the case here).

Another limitation is that the models have not been validated for example by using them to make predictions. One type of prediction could be to fit the model to one biological replica and then predict the other replica (cross validation). Another prediction would be to take the distribution fitted to the experimental data and then compare the model mean to the experimental mean.

Thank you for this comment. As explained above, we used the state-of-the-art PSIS-LOO to measure the predictive performance of the models, which approximates the result of leave-one-out cross-validation using the full data set. To further assess the predictive capabilities of the model, we have now also added a “leave-replicate-out” cross-validation, as the reviewer suggests (new Supp Figure 12). The aim of our “leave-replicate-out” cross-validation was to test how well the predictions of each model generalise to independent cells that are not in the training set. To do this, we trained each model while omitting the data from one gene on a test slide. We then calculated the likelihood score of the test slide using the parameters from the training set, and repeated this for all slides. Similarly to the PSIS-LOO, the results of the leave-replicate-out cross-validation convincingly show that model M4 has the highest predictive performance. This is now described in the updated text on lines 265-271.

The results from fitting and prediction should be plotted as a function of model complexity. This kind of analysis will illustrate how model complexity is supported by the data.

As already mentioned, we used state-of-the-art algorithms to analyze prediction vs. complexity. With the above addition, we now have two methods of calculating the predictive performance of each model: the approximate leave-one-out score as measured with PSIS-LOO and the leave-replicate-out cross-validation. For each model, the PSIS-LOO score is plotted in Figure 3B and the leave-replicate-out cross-validation score is shown in Supp Figure 12.

In the method section on models, a biological motivation must be presented to justify the different model assumption.

Thank you for pointing out that the biological justification of the models needed to be expanded. In addition to the improved justifications already provided in the Results section, we have now updated the Methods section such that a biological motivation is included for each model.

How do the models that fit the distributions describe the mean?

As explained above, the inference is performed on the entire distributions, using a family of distributions (mixtures of NBs) which are parameterized in a biologically relevant manner

(transcriptional bursting + extrinsic noise). The mean and variance of the distribution are now described on lines 585-586 in addition to Figure 3A.

It is necessary to list model parameters for each of the models, their description, their parameter values, their parameter uncertainty and units of each parameter.

Thank you, this has now been added as Supplementary Tables 2-5.

It is not clear to me how the joint probability in figures 2,4, S2 and S4 have been used to fit the model.

Again, the joint distributions are modeled using mixtures of NBs and the inference is performed on the entire dataset at once using a log-likelihood approach. This uses all the data at once, and it is embedded in a Bayesian model selection method. The way that the joint probability is used is now clarified in the revised Methods section and in the Results section (lines 208-214):

“For both models M1 and M2, the likelihood of observing the data given the parameters of the model is evaluated using the model-specific NB distribution and the mRNA counts for both genes in each cell. This is performed for both *Bmal1/Cry1* and *Nr1d1/Cry1* pairs across all time points, and this likelihood is combined with model priors to define the posterior parameter distribution for each model (Methods). We applied Hamiltonian Monte Carlo sampling within the STAN probabilistic programming language to sample the posterior distribution and infer model parameters [40] (parameter estimates for each model shown in Supp Tables 2-5).”

How do the models make sense in the context of the fact that human genes exist as a diploids?

This is a good point, although note though that the 3T3 cells are from mice and not humans. 3T3 cells are tetraploid, and it turns out that under the justified assumption that the bursts are short (Zoller et al., 2015; Suter et al., 2011), the number of alleles rescales the burst frequency, i.e. the effective (observed) burst frequency equals the number of alleles times the burst frequency per allele, but it does not change the shape of the distributions. On line 580-582 we have now written: “Since 3T3 cells are tetraploid, and, again assuming that the bursts are short, the inferred burst frequency for tetraploid cells will be approximately four times that of a single allele.”

The variance decomposition is shortly described but no results are presented to show how this is done. This should be better explained.

The variance decomposition we used is not a new result; in fact, we used the analytical results of Bowsher, C. G. & Swain, P. S. “Identifying sources of variation and the flow of information in biochemical networks” (PNAS, 2012). The mathematical proofs of the formula we use are contained within that reference; however, we have re-written this section to make it clearer to the reader (lines 688-718).

****Minor comments:****

In figure 3A, it is not clear to me what these different plots relate to the models. It is also not clear what are equations that describe each model.

The Methods section has now been improved to show the full data-generating mechanism for each model, and each model has its own section title to make it easier to find. We have also improved the legend for Figure 3 to make the relationship to each model clearer.

The legends in figure 3 are not very informative. More details need to be presented to understand this figure.

Thank you for pointing this out, and we have now re-written the figure legend for Figure 3 to make the figure clearer.

Reviewer #2 (Significance (Required)):

This is an interesting and important topic with the potential to have general implication of how to model periodic single cell gene expression data and eventually better understand circadian clocks. This study will expand on other modeling studies of circadian clocks and has the potential to advance the field (PMCID: PMC7229691). I personally have done similar analysis and experiments in another system and biological context which has demonstrated the power of this approach if implemented rigorously. I am not an expert in circadian clocks in human cells.

We thank the reviewer for appreciating the implications for the circadian and single cell gene expression community. Note that to our knowledge, modeling smFISH counts using mixtures of negative binomials combined with Bayesian model selection has not been done. It is both highly relevant biologically (combines intrinsic and extrinsic fluctuations in a rigorous way), general and its applicability extends far beyond the circadian oscillator. Therefore, this approach for quantitative smFISH data analysis also fills an important methodological gap.

****Referees Cross commenting****

Reviewer #1:

I agree with the assessment that model fitting and model selection was not sufficient. But I disagreed that the data is enough. Although many cells and time points are analyzed, there is no evidence of how reproducible each mRNA distribution can be measured at each time point. I think reproducibility is key and will also help with the model fitting and identification.

Regarding the point on reproducibility, we have made the following four changes:

1. We have added an independent 4 time-point experiment to show that the oscillatory patterns of the distributions are reproducible (Supp Figure 7).
2. In Figure 1 we now also show the mean of each replicate for the main experiment (Figure 1B).
3. We also show the mRNA distributions of each replicate in Supp Figures 2-5.

4. We have added the “leave-replicate-out” cross-validation to show that the model performance of the preferred model generalises to independent slides that were not included in training (Supp Figure 12).

In responding to Reviewer #1 regarding the modeling, we have now also added a simplified dynamical model of circadian clock expression to add mechanistic insight into our proposed models. Overall, we have significantly expanded the description of the model selection approaches to help readers who are less familiar with Bayesian model selection methods.

Reviewer #3:

Regarding the red background, my understanding is that this comes from the probe hybridization. This is maybe because the probe concentration has not been optimized or the number of probes per gene is low and the signal to noise is not so good. Or it could be auto fluorescent background. In this case a different fluorophore needs to be used to avoid this problem.

Thank you for those comments, and we agree with all reviewers that the presentation of the images needed to be improved. It turned out that in Figure 1, we had shown the cell mask in red so it is clearly not related to probe concentration or autofluorescence. We have now removed the cell mask channel from the main images which allows highlighting better the smFISH signals. All smFISH images for Figures 1 and 2 have been much improved, and we've added a new Supp Figure 1 to show the performance of our cell segmentation.

Reviewer #3 (Evidence, reproducibility and clarity (Required)):

In this paper Nicholas et al image mRNAs encoding the key controllers of circadian rhythms, Rev-erba, Cry and Bmal1 in single cells over time. It was shown earlier that single cells exhibit circadian rhythms using reporter genes. A large number of studies have shown that transcription is an inherently stochastic process, which raises a question as to how single cells are able to achieve their rhythms on the face of this noise. Their results show that the number of mRNAs for the three genes exhibit the expected periodicity, but this periodicity is associated with significant cell-to-cell variation. They also explore to what extent this variability derives from stochastic transcription vs other sources of variation that are extrinsic to the genes. The results are interesting and experimental and modeling results are important (however this reviewer is not able to judge the veracity of mathematics that underlay the models).

We thank this reviewer for appreciating the importance of our work.

****Some of the concerns that arose are listed below:****

1. The images show an annoying red background. If the red is HCS cell mask, it should be removed, and RNA presented on grey scale. This will make a better presentation. The red hue also appears in fig 2 b but here it is one of the RNA. I suggest in Fig 2 one RNA can be presented in green and the other in red, while the nuclei in blue.

Thank you for this comment. We had indeed shown the cell mask in the red channel and now removed it. Together with the other suggestions and comments from the reviewers, we implemented the following changes: 1) added the cell contours as requested; 2) use red/green for the smFISH signal in the pairs of genes; 3) we have improved the contrast to make it easier to distinguish the RNA FISH signals. The presentation of the images is now much improved.

2. This paper and a few others talk about the cell size contributing to the cell-to-cell variability in mRNA numbers. Where does it come from physically? One can imagine based on the cell cycle stage there could be more than two copies of then gene in a cell, which will yield more RNAs, but they say that their cells don't have much cell cycle variability. Perhaps a clearer discussion is called for rather than just being polite to other investigators.

The referee is right that several studies observed empirically that larger cells show more mRNA molecules in smFISH experiments (Padovan et al., 2015; Kempe et al., 2015). In Padovan et al. (2015), the authors found that transcriptional burst size changes with cell volume and burst frequency with cell cycle. The main theory for transcription scaling with cell volume is to maintain transcript concentration. Using cell fusion experiments, they showed that cellular size can directly and globally affect gene expression by modulating transcription. Furthermore, they proposed that the mechanism underlying the global regulation integrates both DNA content and cellular volume to produce the appropriate amount of RNA for a cell of a given size, which is consistent with a model whereby a factor limiting for transcription is sequestered to the DNA. We used these results to propose a model whereby burst size scales with area, and we found an increase in predictive performance (compare M2 with M1 in Figure 3B). While our model selection supported the inclusion of cell area, the variance decomposition showed that the fraction of variance due to cell area ranged from 4.2% for *Nr1d1* to 17.6% for *Bmal1*. We have now expanded the introduction to discuss this in more depth (lines 73-80) as requested.

3. References 26 and 27 are cited for 10-80% of variance due to gene extrinsic sources. These references actually deny that there is a significant transcriptional noise in most genes. Again, stronger discussion is called for.

As mentioned in the reply to Reviewer 1, previous work from our lab is also nuancing the conclusions from references 26 and 27. Specifically, buffering effects are expected to be highly gene-specific (3'UTR), and in fact we have not seen those with our unstable construct during live-cell imaging (Suter et al., 2011; Zoller et al., 2015). We have also added text in order to explicitly state that subsequent papers have nuanced the general claims in references 26 and 27. In the text we write (lines 335-342):

“One explanation for the low intrinsic fluctuation in these studies is that transcriptional fluctuations are filtered by nuclear retention, though other reports suggest that Fano factors (variance/mean, a measure of overdispersion compared to the Poisson distribution) can be even larger in the cytoplasm than in the nucleus [38]. In the cells used here, the strong signature of transcriptional bursting and high intrinsic noise is consistent with live imaging of a *Bmal1* transcriptional reporter in the same cell line under similar growth conditions, where intrinsic noise was estimated to be 4-times larger than extrinsic noise [23].”

4. The results raise a very important question, whether and to what extent the transcriptional noise propagates to the next step of gene regulation and are there buffering mechanisms in the cell. For example, Raj et al, *Variability in gene expression underlies incomplete penetrance*, *Nature* 2010, show that alternative pathways serve to buffer the impact of gene expression noise. Similarly, Shah and Tyagi, *Barriers to transmission of transcriptional noise in a c-fos c-jun pathway*, *Mol Syst Biol*, 2013, show that variability in mRNA is buffered at protein level and the level of protein-protein complexes. Furthermore, they show that to the extent those vary, the chromatin intrinsically buffers against the fluctuations in numbers of transcription factors. Mention of these and other studies will enrich the paper.

We have modified the Discussion section and now discuss these papers (and a few more). We thank the reviewer for the suggestions, which will help the reader to have a broader overview of noise buffering in gene expression and indeed enrich the paper.

Reviewer #3 (Significance (Required)):

Significance is high. Quality is high.

****Referees Cross-Commenting****

I agree with the comments made by other reviewers particularly about references 26 and 27. The major conclusions of reference 26 were questioned by Hansen et al 2018. At the bottom of page 7 the authors are qualifying their results in the light of references 26 and 27. Perhaps now there is less of a need to do so.

As mentioned above, we have added the following sentence citing the Hansen paper to make it clear to the reader that key conclusions of the references 26 and 27 are disputed (lines 335-342):

“One explanation for the low intrinsic fluctuation in these studies is that transcriptional fluctuations are filtered by nuclear retention, though other reports suggest that Fano factors (variance/mean, a measure of overdispersion compared to the Poisson distribution) can be even larger in the cytoplasm than in the nucleus [38].

References

- Gelman A, Carlin JB, Stern HS, Dunson DB, Vehtari A, Rubin DB. 2013. *Bayesian Data Analysis*, 3rd edn. CRC Press, London.
- Hughes ME, DiTacchio L, Hayes KR, Vollmers C, Pulivarthy S, Baggs JE, Panda S, Hogenesch JB. 2009. Harmonics of circadian gene transcription in mammals. *PLoS Genet* **5**. doi:10.1371/journal.pgen.1000442
- Kempe H, Schwabe A, Cremazy F, Verschure PJ, Bruggeman FJ. 2015. The volumes and transcript counts of single cells reveal concentration homeostasis and capture biological noise. *Mol Biol Cell* **26**:797–804. doi:10.1091/mbc.E14-08-1296

- Padovan-Merhar O, Nair GP, Biaesch AG, Mayer A, Scarfone S, Foley SW, Wu AR, Churchman LS, Singh A, Raj A. 2015. Single Mammalian Cells Compensate for Differences in Cellular Volume and DNA Copy Number through Independent Global Transcriptional Mechanisms. *Mol Cell* **58**:339–352. doi:10.1016/j.molcel.2015.03.005
- Raj A, Peskin CS, Tranchina D, Vargas DY, Tyagi S. 2006. Stochastic mRNA synthesis in mammalian cells. *PLoS Biol* **4**:e309. doi:10.1371/journal.pbio.0040309
- Relógio A, Westermark PO, Wallach T, Schellenberg K, Kramer A, Herzog H. 2011. Tuning the mammalian circadian clock: Robust synergy of two loops. *PLoS Comput Biol* **7**:1–18. doi:10.1371/journal.pcbi.1002309
- Saini C, Morf J, Stratmann M, Gos P, Schibler U. 2012. Simulated body temperature rhythms reveal the phase-shifting behavior and plasticity of mammalian circadian oscillators. *Genes Dev* **26**:567–580. doi:10.1101/gad.183251.111
- Suter DM, Molina N, Gatfield D, Schneider K, Schibler U, Naef F. 2011. Mammalian Genes Are Transcribed with Widely Different Bursting Kinetics. *Science (80-)* **332**:472–474. doi:10.1126/science.1198817
- Ukai-Tadenuma M, Yamada RG, Xu H, Ripperger JA, Liu AC, Ueda HR. 2011. Delay in feedback repression by cryptochrome 1 Is required for circadian clock function. *Cell* **144**:268–281. doi:10.1016/j.cell.2010.12.019
- Vehtari A, Gelman A, Gabry J. 2017. Practical Bayesian model evaluation using leave-one-out cross-validation and WAIC. *Stat Comput* **27**:1413–1432. doi:10.1007/s11222-016-9696-4
- Wu C, Simonetti M, Rossell C, Mignardi M, Mirzazadeh R, Annaratone L, Marchiò C, Sapino A, Bienko M, Crosetto N, Nilsson M. 2018. RollFISH achieves robust quantification of single-molecule RNA biomarkers in paraffin-embedded tumor tissue samples. *Commun Biol* **1**:1–8. doi:10.1038/s42003-018-0218-0
- Zoller B, Nicolas D, Molina N, Naef F. 2015. Structure of silent transcription intervals and noise characteristics of mammalian genes. *Mol Syst Biol* **11**:823. doi:10.15252/msb.20156257

Thank you for submitting your work to Molecular Systems Biology along with the referee reports from Review Commons. We have now heard back from reviewer #2 who agreed to evaluate your revised study. As you will see below, reviewer #2 is satisfied with the performed revisions and is supportive of publication in Molecular Systems Biology.

Before we can formally accept the study for publication, we would ask you to address the following editorial issues.

Reviewer #2:

The authors have answered my criticism fully and satisfactorily. The manuscript can be accepted for publication.

1st Authors' Response to Reviewers**5th Jan 2021**

The authors have addressed the remaining editorial issues.

Accepted**19th Jan 2021**

Thank you for performing the requested edits. We are now satisfied with the modifications made and I am pleased to inform you that your paper has been accepted for publication.

Corresponding Author Name: Felix Naef

Manuscript Number: MSB-2020-10135